# Topological turning points across the human lifespan

Alexa Mousley [1] ✉, Richard A. I. Bethlehem [2], Fang-Cheng Yeh [3] & Duncan E. Astle [1,4]

Structural topology develops non-linearly across the lifespan and is strongly related to cognitive trajectories. We gathered diffusion imaging from datasets with a collective age range of zero to 90 years old ($N$ = 4,216). We analyzed how 12 graph theory metrics of organization change with age and projected these data into manifold spaces using Uniform Manifold Projection and Approximation. With these manifolds, we identified four major topological turning points across the lifespan – around nine, 32, 66, and 83 years old. These ages defined five major epochs of topological development, each with distinctive age-related changes in topology. These lifespan epochs each have a distinct direction of topological development and specific changes in the organizational properties driving the age-topology relationship. This study underscores the complex, non-linear nature of human development, with unique phases of topological maturation, which can only be illuminated with a multivariate, lifespan, population-level perspective.

Trajectories of change in brain structure and function emerge across the lifespan[1–4]. Topology, the complex motifs within which neural connections are organized, develops with age and is associated with key cognitive, behavioral, and mental health outcomes[5–10]. Topology-outcome relationships have been established within relatively narrow age ranges, such as childhood[6,7,10]. But what are the underlying *principles* of organizational change? Are there key points in our lifespans wherein the brain transitions into a different phase of developmental change? Addressing these questions requires comprehensive mapping of lifespan network topology alongside a multidimensional framework capable of establishing the non-linear dynamics of developmental change.

Prior research has revealed significant differences in structural topology associated with both individual differences[6,9–11] and lifespan development[4,12–15]. A typically developing infant's brain network displays adult-like structure with hub distribution, rich clubs, small-worldness, and modularity at birth[16–25]. Throughout early development, networks become more integrated with increasing strength and efficiency and decreasing modularity[26–28]. In adulthood, many researchers describe an inverted "U" shape of development with a peak

occurring around 30 years old where the brain is maximally efficient and integrated[13–15]. This research uses the terms *inflection point*[1,2,29,30] or *peak age*[14,15] to describe important points of change in organizational metrics – many of which occur in the fourth decade of life and intersect with other developmental and aging milestones. After this point and into late life, aging is associated with reduced connectivity, mainly through pruning of weak connections[12,14,31], increased modularity[13], and more pronounced rich club organization[14] than earlier in life. In addition to these age-related changes, topological variation is associated with differences in individual outcomes. For example, there is a positive association between global efficiency (more short paths for information transfer) and intelligence in children[7] and a negative association between global efficiency and cognitive impairments in aging individuals[11]. These established variations and lifespan fluctuations of organizational principles underscore the dynamic and complex nature of topology development.

Mapping neural systems across the lifespan calls for data-driven methods that can handle complex data and capture high variability without making strong assumptions about the underlying data[32]. Manifold learning is a popular technique to project high-dimensional

[1]MRC Cognition and Brain Sciences Unit, University of Cambridge, Cambridge, UK. [2]Department of Psychology, University of Cambridge, Cambridge, UK. [3]Department of Neurological Surgery, University of Pittsburgh, Pittsburgh, PA, USA. [4]Department of Psychiatry, University of Cambridge, Cambridge, UK. ✉ e-mail: alexa.mousley@mrc-cbu.cam.ac.uk

data into a low-dimensional space by filtering out closely related features likely driven by similar mechanisms[33,34]. These projections, created using non-linear dimensionality reduction techniques such as Uniform Manifold Approximation and Projection (UMAP), are easier to interpret while maintaining the intrinsic structure of complex data[33,35]. UMAP, in particular, captures both local and global data patterns with faster run times than similar methods (e.g., t-SNE)[35]. The manifold spaces themselves graphically represent the relationships in the original high-dimensional space and, therefore, offer an opportunity to leverage projections to identify points of change. We define these points of change as *turning points*, indicating significant shifts in the overall trajectory of topology, rather than inflection or peak points, which refer to changes in individual organizational measures.

This study takes a data-driven approach to chart structural topological development across the human lifespan. Specifically, we (1) characterize connectivity development; (2) explore topological integration, segregation, and centrality; (3) use UMAP to define topological manifold spaces and identify major turning points across the lifespan therein; and (4) examine how these turning points capture important phases of topological development.

## Results

We gathered diffusion imaging data from nine datasets with a combined age range of zero to 90 years old (Fig. 1a, b; Supplementary Table 1). A large sample ($N = 4216$) including all available images was fiber tracked[36] and harmonized[37] (Fig. 1c). For analysis, multiple graph theory metrics[38] (Supplementary Table 2) were calculated using normalized weighted networks with a cross-sectional, neurotypical subset ($n = 3802$; female $n = 1994$; male $n = 1808$). Following topological analysis, we projected age-predicted organizational measures into

manifold spaces using UMAP[35] and determined significant turning points in topological development across the lifespan.

### Connectivity

Before exploring network organization, we first examined general changes in connectivity across the lifespan by preserving the distribution of density by applying an absolute streamline-count threshold that yielded densities that were 70% of the average raw density per single-year age bin (Fig. 1c; Supplementary Fig. 1a). Density—the percent of connections present in the network[38]—changed non-linearly across age with high-density networks present around birth and 30 years old and sparse networks observed around 10 and 80+ years old (Fig. 2b; $F_{density,age} = 219.20$, estimated df = 8.92, $p < 2.00 \times 10^{-16}$). In addition, node strength—the sum of edge weights[38]—significantly increased across the lifespan in a linear pattern both in average strength across all nodes and maximum strength across all nodes (Fig. 2c; $F_{average\ strength,age} = 33.10$, estimated df = 5.46, $p < 2.00 \times 10^{-16}$; $F_{maximum\ strength,age} = 29.15$, estimated df = 3.85, $p < 2.00 \times 10^{-16}$). Overall, these networks displayed the expected pattern of shifting from dense, weak networks in early life to sparse, strong networks in later life[20,39] (Fig. 2a).

### Topology

To remove the confounding factor of network density from the analysis of topological changes, we conducted a density-controlled analysis where each network was constrained to exactly 10% density (Fig. 1c). This method allows for fair comparison of topological structure across the lifespan without total connectivity biases, though a full topological analysis with variable density networks is also provided (Supplementary Fig. 2, Table 3). Topological metrics can be

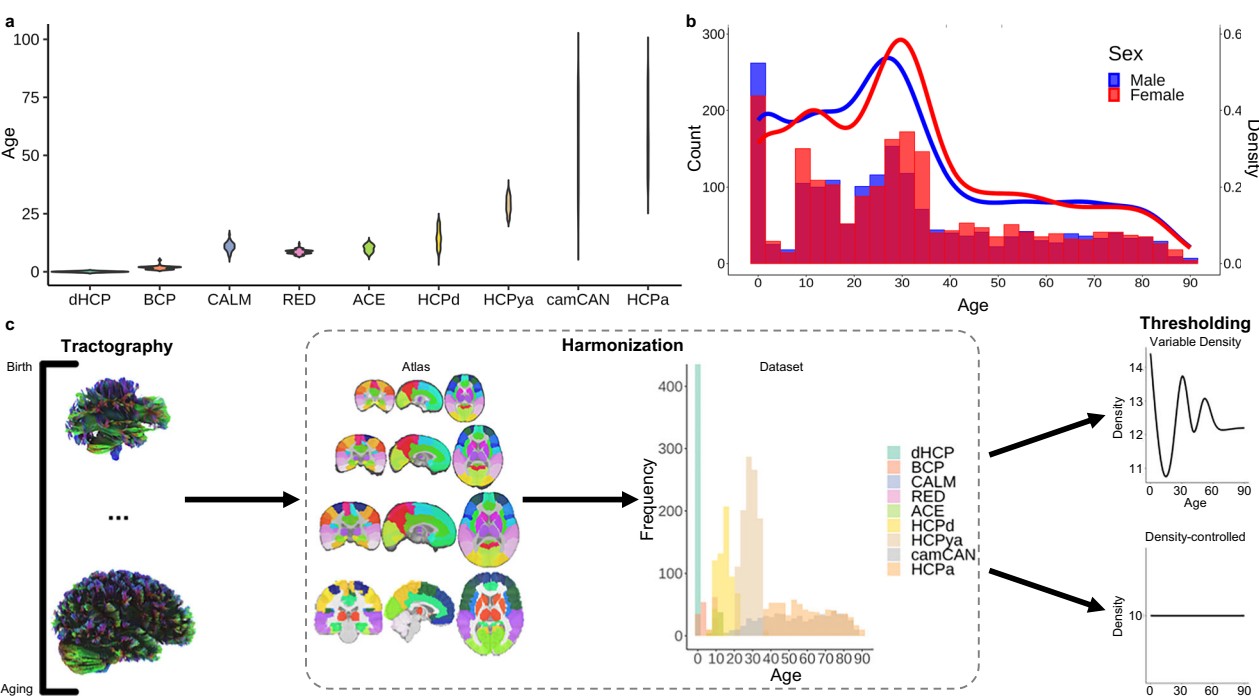

**Fig. 1 | Datasets demographics, methods schematic, and network connectivity. a** The distribution of ages (years) across each dataset (dHCP = Developing Human Connectome Project; BCP = Baby Connectome Project; CALM = Centre for Attention Learning and Memory; RED = Resilience in Education and Development; ACE = Attention and Cognition in Education; HCPd = Human Connectome Project Development; HCPya = Human Connectome Project Young Adult; camCAN = Cambridge Centre for Ageing and Neuroscience; HCPa = Human Connectome Project Ageing). **b** A histogram and density plot of sex distribution across age for the entire sample. **c** Methods schematic demonstrating that fiber tracking was

performed for all participants, each registered to an age-appropriate AAL90 atlas, followed by harmonization using the ComBat algorithm[37] across the atlases and datasets. Next, two thresholding analyses were conducted—variable density and density-controlled. For variable density analysis, networks were thresholded to an age-specific average density (70% of raw density) to preserve variation in network density across the lifespan. For the density-controlled analysis, all networks were thresholded to exactly 10% density[91] to allow for direct topological comparisons, which are not biased by differences in total connectivity.

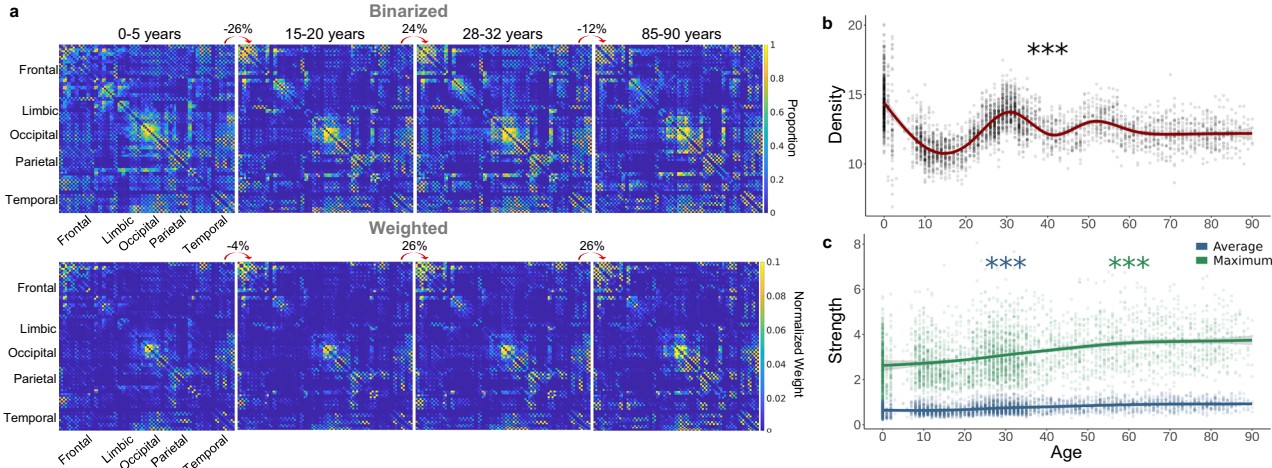

**Fig. 2 | Changes in total network connectivity across the lifespan. a** Average binarized connectivity matrices and average normalized weighted connectivity matrices. Above each consecutive pair of matrices, the percent change indicates the difference in total connectivity. **b** Density (%) significantly fluctuated across the lifespan with a lifetime maximum at birth and minimum around 14 years old ($p < 2.00 \times 10^{-16}$) (assessed via GAM). (**c**) The maximum ($p < 2.00 \times 10^{-16}$) and average strength ($p < 2.00 \times 10^{-16}$) significantly and nearly linearly increased across the lifespan with a lifetime minimum at eight years old and maximum at 90 years old (assessed via GAM). Shaded area around lines of best fit represents 95% confidence intervals. *** indicates $p < 0.001$, ** indicates $p < 0.01$, * indicates $p < 0.05$.

categorized as measures of network integration, segregation or centrality. Integration measures, such as global efficiency, assess the ease of communication across the network[38]. Highly integrated topology is typically achieved by the network being well-connected by short path lengths, which conveys that the network is optimized for efficient communication[38]. Network segregation, on the other hand, relates to partitioning the network into subgroups (e.g., modules), which are typically measured through the density or strength of within-group connections[38]. Segregated topology increases the network's capacity for specialized processing by use of subunits within a larger complex network[38,40]. Lastly, measures of centrality convey the presence of nodes that are particularly important for network function (i.e., 'central' nodes)[38]. For example, a node that is a member of numerous shortest paths is highly central as it plays a key role in information transfer. Thus, centrality not only facilitates network communication but also increases networks' resilience to random knockouts of nodes[41]. All these concepts are important to understand in the context of the lifespan because different topological structures have strengths and weaknesses related to network function and thus provide clues as to the 'goals' of developmental change.

Global efficiency, which measures how well the network is connected by short path lengths[42], significantly fluctuated across the lifespan, peaking at 29 years old before steadily declining to a minimum at 90 years old (Fig. 3a; Table 1). Other integration metrics include characteristic path length, the average shortest path length of the network[43], and small-worldness, the ratio of the average clustering coefficient to characteristic path length[44]. Both showed similar but inverse patterns to global efficiency (Fig. 3a; Table 1). Additionally, average network strength significantly increased in a more linear pattern, reaching its maximum at 90 years old (Fig. 3a; Table 1). These results suggest that while network strength linearly increases with age, topological integration initially decreases in the first decade, peaks at the beginning of the fourth decade, and then declines for the rest of the lifespan.

Modularity, how well a network can be divided into non-overlapping, highly intra-connected node groups[45], significantly fluctuated across the lifespan with a minimum at 31 years old and a maximum at 90 years old (Fig. 3b; Table 1). Core/periphery structure, which assesses how well a network separates into a non-overlapping dense core and a sparse periphery[38], fluctuated more than modularity, peaking at 20 years old and reaching a minimum at 55 years

old (Fig. 3b; Table 1). Additionally, networks can be segregated based on a subnetwork with a specific strength (i.e., s-core) or degree (i.e., k-core)[38]. While k-core did not significantly change across age, s-core significantly fluctuated across the lifespan with a minimum at 12 years old, followed by a continuous increase to a maximum at 90 years old.

Compared to global segregation metrics, average local segregation measures increased more linearly across the lifespan. Local efficiency—the extent to which neighboring nodes are connected by short paths[42]—and clustering coefficient—the extent to which neighboring connected nodes are also connected to each other[43]—both significantly increased to a maximum at 90 years old (Fig. 3b; Table 1). These results emphasize a difference between global segregation, which oscillated across age, and average local segregation, which showed more linear patterns. Beyond differences in fluctuations in mid-life, network segregation peaked in late life.

Centrality measures a node's importance to the network, often based on inclusion in key paths. Betweenness centrality measures the fraction of shortest path lengths that pass through the node[46], which fluctuated across the lifespan, reaching a minimum at 31 years old and maximum at 90 years old (Fig. 3c; Table 1). Comparatively, subgraph centrality—the weighted sum of all closed walks for a node[38]—significantly increased in a more linear pattern (Fig. 3c; Table 1). These results highlight differences in the developmental pattern between individual centrality metrics but indicate a continuous increase in centrality starting around the fifth decade.

Generally, network organization displays linear and fluctuating patterns across the lifespan. Various sex effects were found, though these results could be explained by brain size differences that are not considered in this analysis[31,47] (Supplementary Fig. 3). Overall, average strength, average local efficiency, average clustering coefficient, s-core, and subgraph centrality display linear-like patterns while the other metrics appear to have peaks and valleys throughout the lifespan —many of which appear around 30 years old.

## Construction of lifespan epochs

Many topological measures are highly correlated and therefore convey redundant and unique topological characteristics (Supplementary Fig. 4a). Thus, we reduced the dimensionality of this data using manifold learning to examine non-linear changes in lifespan topology. These manifolds are 3-dimensional topological spaces that

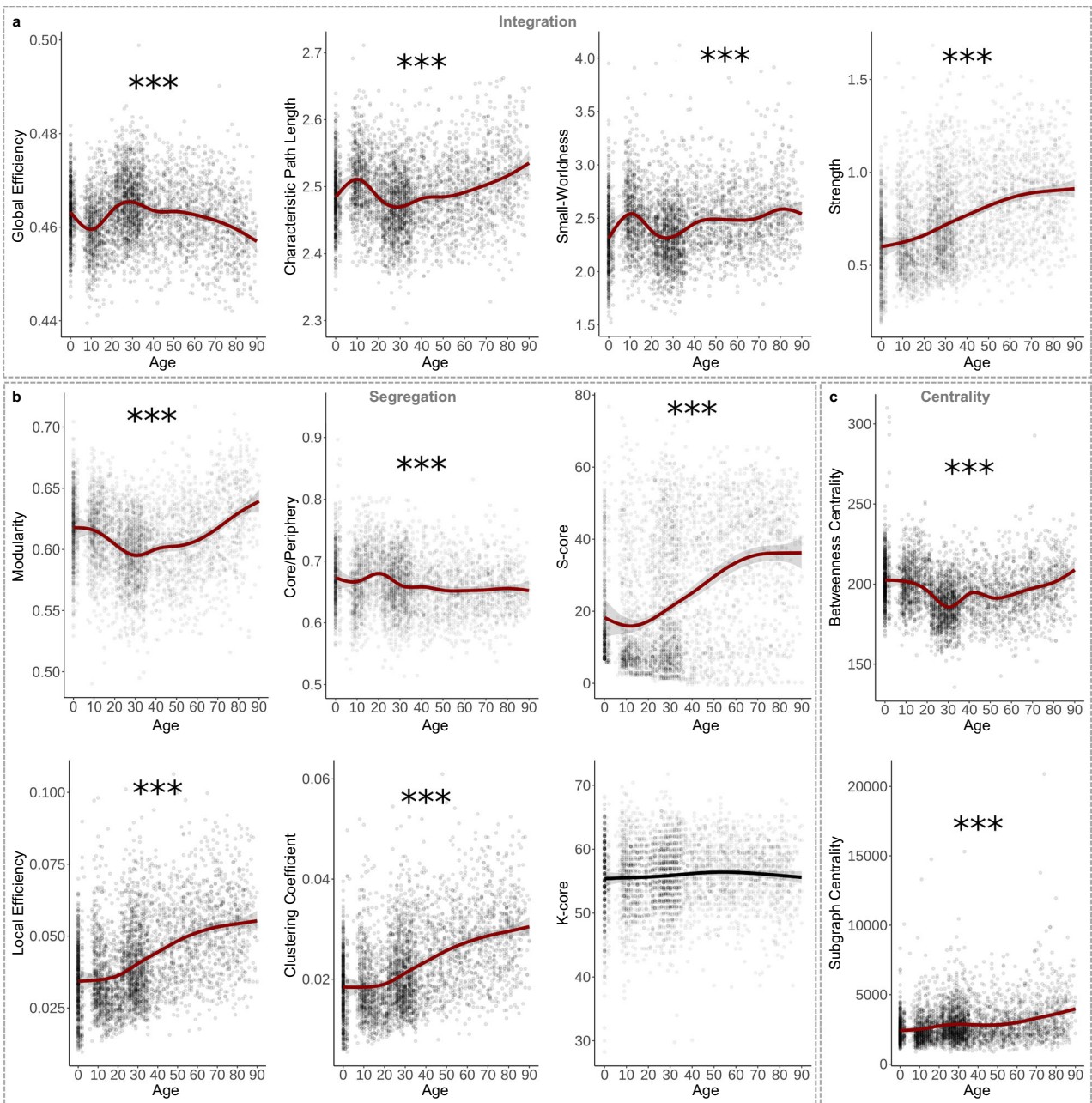

**Fig. 3 | Topological changes across the lifespan in density-controlled networks.** **a** Global efficiency peaked while characteristic path length displayed a lifetime minimum around 29 years old ($p < 2.00 \times 10^{-16}$). Small-worldness ($p < 2.00 \times 10^{-16}$) showed a similar developmental pattern to characteristic path length ($p < 2.00 \times 10^{-16}$), with all ages demonstrating the presence of small-world structure (i.e., small-worldness > 1). Average network strength, however, significantly increased across the lifespan in a more linear pattern ($p < 2.00 \times 10^{-16}$). **b** Modularity significantly fluctuated across the lifespan, peaking in aging individuals ($p < 2.00 \times 10^{-16}$). Core/periphery structure had a lifetime peak around 20 years old ($p < 2.00 \times 10^{-16}$). S-Core (reported as the number of nodes included in the subnetwork) significantly increased across the lifespan in a

linear-like pattern ($p < 2.00 \times 10^{-16}$). Local efficiency ($p < 2.00 \times 10^{-16}$) and clustering coefficient ($p < 2.00 \times 10^{-16}$) both significantly increased linearly across the lifespan. K-Core (reported as the number of nodes included in the subnetwork) did not significantly change across the lifespan ($p = 0.192$). **c** Average betweenness centrality had a lifetime minimum around 31 years old and significantly increased in late life ($p < 2.00 \times 10^{-16}$). Average subgraph centrality significantly increases across the lifespan ($p < 2.00 \times 10^{-16}$). All graphs are GAMs. Shaded area around lines of best fit represents 95% confidence intervals. *** indicates $p < 0.001$, ** indicates $p < 0.01$, * indicates $p < 0.05$.

capture crucial patterns in the data. Manifolds were constructed using significant age-predicted metrics (i.e., excluding k-core), which were averaged for each age. Considering the influence of parameter choice on UMAP projections[48], we created 968 UMAPs with a variety of parameters to capture both local and global-level information (Fig. 4). Manifolds were then used to determine major turning points across the lifespan, marking epochs where topological development is occurring along the same trajectory (Fig. 4c; see "Methods"; "Turning point identification"). Major turning points occur around age nine, 32, 66, and 83 (Fig. 4c). Sex-stratified projections and major turning point are in the supplement (Supplementary Fig. 5). These turning points define five major epochs of life: Epoch One, which lasts from zero to nine years; Epoch Two, which extends from nine to 32 years; Epoch Three, which ranges from 32 to 66 years; Epoch Four, which includes 66 to 83 years; and Epoch Five, which extends from 83 to 90 years.

**Table 1 | Generalized additive model statistics of graph theory metrics across the lifespan**

| | | F-value | Estimated df | p-value | Lifespan Min. | Lifespan Max. | Peaks/ Valleys |
|---|---|---|---|---|---|---|---|
| **Integration** | Global Efficiency | 42.33 | 7.93 | <2.00 × 10⁻¹⁶ | 90 | 29 | 9, 29, 45, 50 |
| | Characteristic Path Length | 43.55 | 7.83 | <2.00 × 10⁻¹⁶ | 29 | 90 | 9, 29, 46, 48 |
| | Small-Worldness | 32.62 | 8.08 | <2.00 × 10⁻¹⁶ | 0 | 82 | 11, 27, 49, 61, 82 |
| | Average Strength | 45.64 | 3.89 | <2.00 × 10⁻¹⁶ | 0 | 90 | - |
| **Segregation** | Modularity | 60.93 | 6.67 | <2.00 × 10⁻¹⁶ | 31 | 90 | 31 |
| | Core/Periphery | 9.04 | 7.60 | <2.00 × 10⁻¹⁶ | 55 | 20 | 8, 20, 35, 40, 55, 80 |
| | S-Core | 43.63 | 5.00 | <2.00 × 10⁻¹⁶ | 12 | 90 | 12 |
| | K-Core | – | – | 0.192 | – | – | – |
| | Local Efficiency | 54.29 | 4.69 | <2.00 × 10⁻¹⁶ | 0 | 90 | - |
| | Clustering Coefficient | 62.42 | 4.76 | <2.00 × 10⁻¹⁶ | 6 | 90 | 6 |
| **Centrality** | Betweenness Centrality | 55.29 | 8.29 | <2.00 × 10⁻¹⁶ | 31 | 90 | 30, 42, 52 |
| | Subgraph Centrality | 55.29 | 4.86 | <2.00 × 10⁻¹⁶ | 0 | 90 | 30, 44 |

Lifespan minimum (Min.), Lifespan maximum (Max.), and Peaks/Valleys are reported as age in years. Estimated degrees of freedom is a GAM output indicating the model's complexity or 'wiggliness'. Peaks/Valleys are ages at which the derivative of the GAM switches sign (–/+ or +/–).

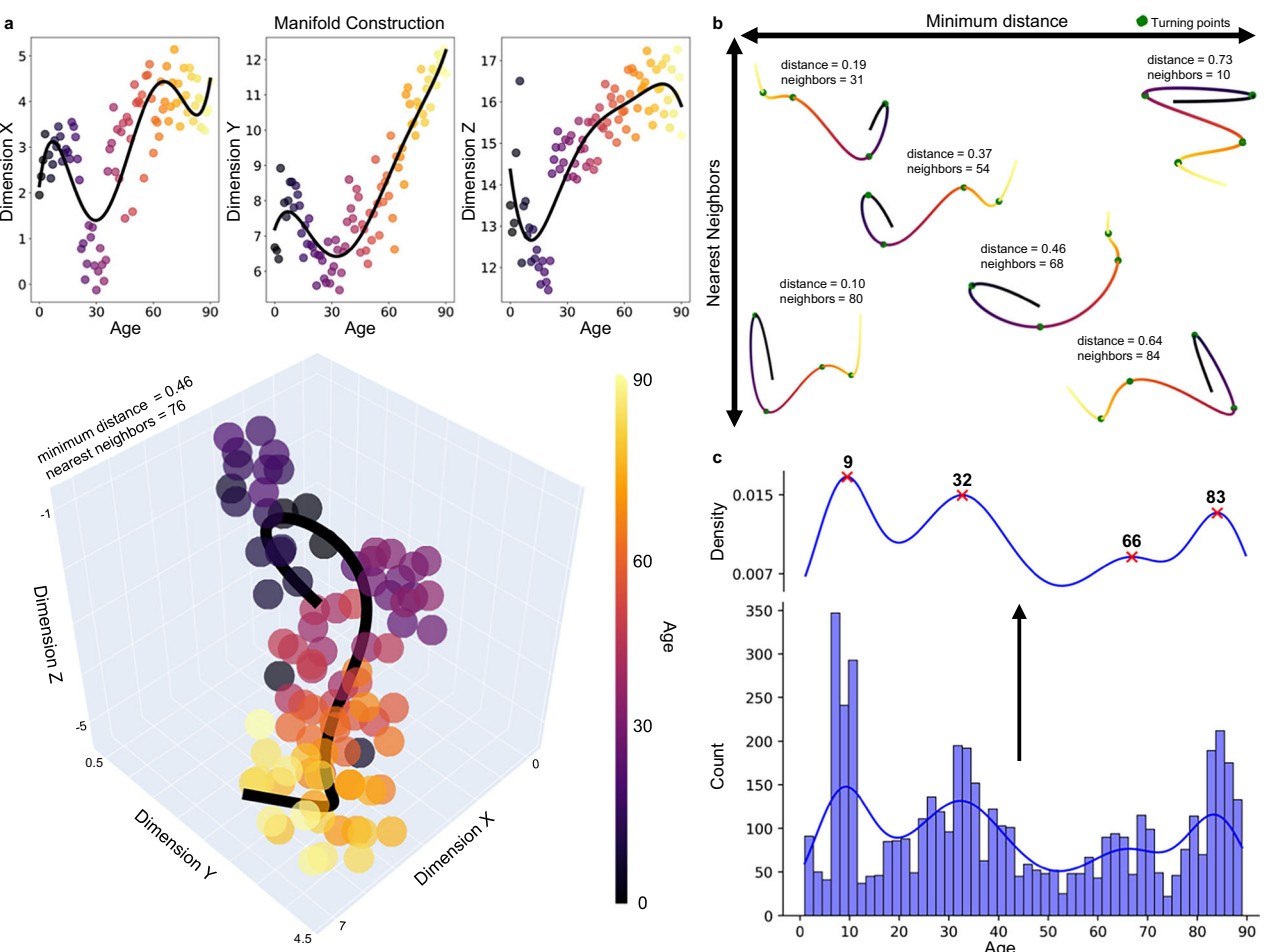

**Fig. 4 | The definition of turning points. a** A manifold space displayed across each dimension (top row) and in a 3-dimensional plot. The scatter plots show the age-average UMAP projection with the polynomial lines of best fit (black). Lines of best fit were constructed for each of the 968 UMAPs. **b** Six examples of lines of best fit through manifolds with different UMAP parameters. These lines are used to determine turning points (green points). **c** All turning points identified across the 968 UMAP projections are plotted in a histogram and kernel density plot. These plots were used to determine the major turning points, which are ages most frequently identified as turning points across all projections. The major turning points occur at nine, 32, 66, and 83 years old (red 'x').

We explored changes across these epochs using Pearson correlations to assess *directional* relationships between age and topological measures and LASSO regularized regressions to identify which organizational properties *drive* the relationship between topology and age.

At each turning point, we analyzed significant changes in directionality and key driving topological metrics. Local-level correlations between measures and age within each epoch are in the supplement (Supplementary Fig. 6).

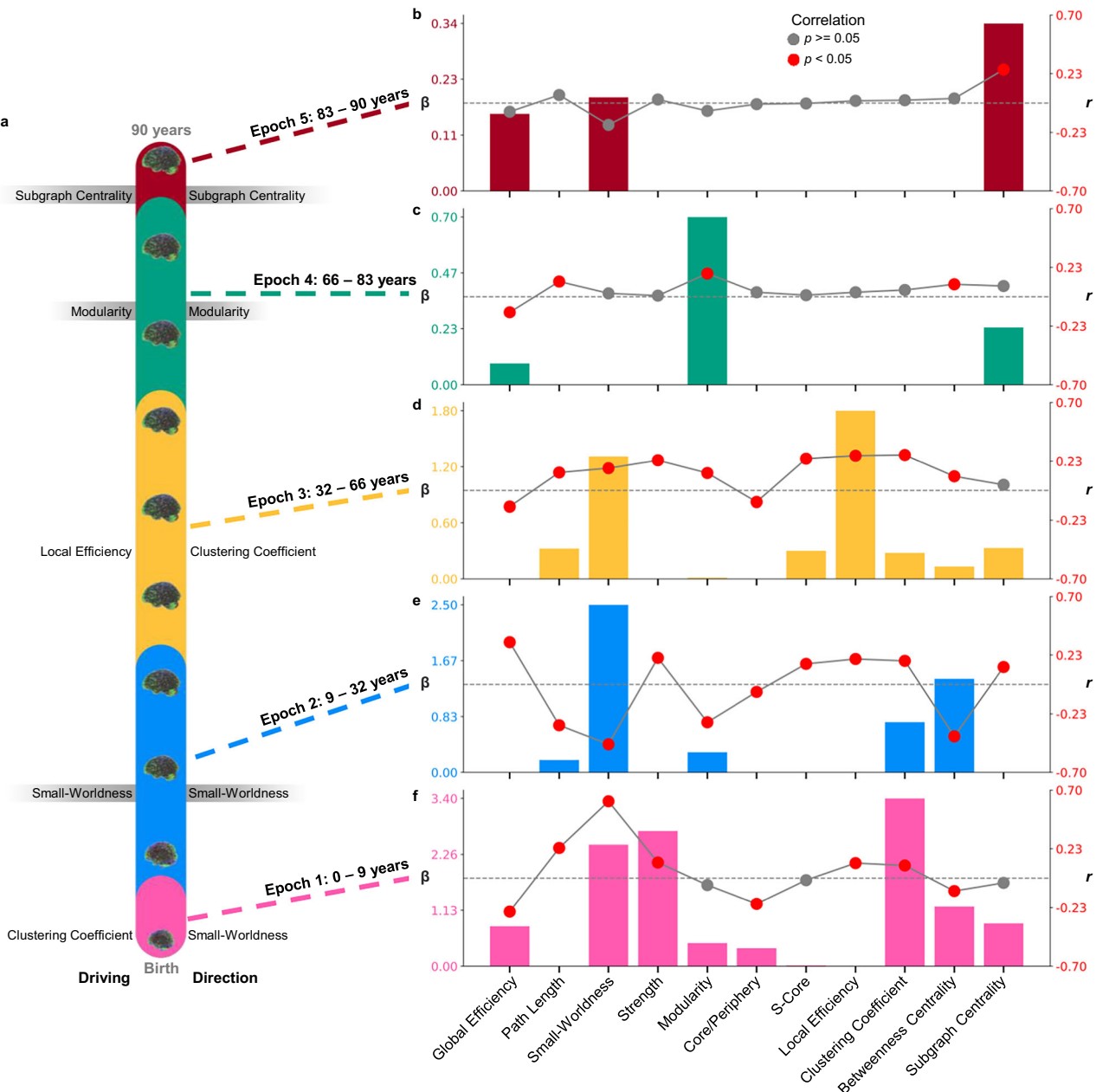

**Fig. 5 | Topological changes within the five topological epochs of life.**
**a** Schematic showing the ranges of each epoch across the lifespan. To the left of the schematic, per epoch, the metric with the highest LASSO coefficient (β), indicating the strongest predictor of age, is shown. On the right side of the schematic, next to each epoch, the metric with the largest correlation (r) is displayed, highlighting the strongest directional changes across age. For the epochs in which the driving factor and strongest directional change are the same, the metrics are highlighted in gray. The combination plots show all the regularized LASSO coefficients in a bar graph (left y-axis) and the correlations as a scatter plot (right y-axis) for (**b**) epoch 5, (**c**) epoch 4, (**d**) epoch 3, (**e**) epoch 2, and (**f**) epoch 1. Red scatter points indicate Pearson correlations with $p < 0.05$. Dotted gray lines highlight zero on the correlation axis.

## Epoch 1: 0–9 years old "infancy into childhood"

The first epoch ranges from zero to nine years ($n = 733$), covering the period of infancy through childhood. Significant correlations were found within this epoch in eight organizational measures, characterized by decreasing global integration, increasing local segregation, and stable centrality (Fig. 5f; Table 2). The regularized LASSO regression retained eight measures and identified the clustering coefficient as the strongest topological predictor of age ($\lambda = 0.04$; Fig. 5f; Table 2). In contrast, small-worldness has the largest correlation with age in this epoch (Fig. 5f; Table 2). Thus, while an increase in small-worldness across this period is the largest *directional* pattern, the average clustering coefficient is the crucial predictor of age (Fig. 5a). Significant correlations between clustering and age were found in 55 out of the 90

regions after false-discovery rate (FDR) correction, and these regions are generally disrupted across the brain (Supplementary Fig. 6c). Overall, topological development from zero to nine years old is characterized by decreasing global integration, however, clustering coefficient is a key topological measure across this period. Thus, despite decreasing integration overall, a child's age is most distinct topologically in the extent to which neighboring nodes are interconnected.

The first epoch of life ends around nine years old, which was the most frequently identified turning point, occurring 241 times across all UMAPs (Fig. 4c). Around nine years old, we observed the factor driving the relationship between topology and age shift from clustering coefficient to small-worldness (Fig. 5a). Directional changes occur as

**Table 2 | Regularized LASSO coefficients and correlations within epochs of life**

| | | Epoch 1 | Epoch 2 | Epoch 3 | Epoch 4 | Epoch 5 |
|---|---|---|---|---|---|---|
| **Integration** | Global Efficiency | r = −0.27*** | r = 0.34*** | r = -0.13*** | r = −0.12* | – |
| | | β = 0.81 | – | – | β = 0.09 | β = 0.16 |
| | Characteristic Path Length | r = 0.24*** | r = −0.32*** | r = 0.14*** | r = 0.12* | – |
| | | – | β = 0.19 | β = 0.32 | – | – |
| | Small-Worldness | r = 0.61*** | r = −0.47*** | r = 0.17*** | – | – |
| | | β = 2.46 | β = 2.50 | β = 1.31 | – | β = 0.19 |
| | Strength | r = 0.13*** | r = 0.21*** | r = 0.24*** | – | – |
| | | β = 2.74 | – | – | – | – |
| **Segregation** | Modularity | – | r = −0.30*** | r = 0.14*** | r = 0.18*** | – |
| | | β = 0.47 | β = 0.30 | β = 0.01 | β = 0.70 | – |
| | Core/Periphery | r = −0.20*** | r = −0.06* | r = −0.09** | – | – |
| | | β = 0.36 | – | – | – | – |
| | S-Core | – | r = 0.16*** | r = 0.25*** | – | – |
| | | β = 0.02 | – | β = 0.30 | – | – |
| | Local Efficiency | r = 0.12** | r = 0.20*** | r = 0.28*** | – | – |
| | | – | – | β = 1.80 | – | – |
| | Clustering Coefficient | r = 0.10** | r = 0.19*** | r = 0.28*** | – | – |
| | | β = 3.40 | β = 0.75 | β = 0.28 | – | – |
| **Centrality** | Betweenness Centrality | r = −0.10** | r = −0.41*** | r = 0.11*** | r = 0.10* | – |
| | | β = 1.21 | β = 1.40 | β = 0.13 | – | – |
| | Subgraph Centrality | – | r = 0.14*** | r = 0.05* | r = 0.09* | r = 0.27* |
| | | β = 0.87 | – | β = 0.33 | β = 0.24 | β = 0.34 |

r is the Pearson correlation coefficient and β is the LASSO coefficient. Dotted lines indicate where the direction of significant correlations between consecutive epochs occurs. Dashes indicate a non-significant correlation or zero β coefficients. *** indicates $p < 0.001$, ** indicates $p < 0.01$, * indicates $p < 0.05$. All correlation *p-values* are located in Supplementary Table 4.

well, with significantly decreasing integration changing to significantly increasing integration after nine years old (Table 2; Supplementary Fig. 7a).

### Epoch 2: 9–32 years old "Adolescence"
The second epoch occurs from nine to 32 years old ($n = 1,728$) and encompasses late childhood through early adulthood. Within this epoch, all topological measures were significantly correlated with age, characterized by increasing network integration and complex segregation and centrality patterns (Fig. 5e; Table 2). Generally, strength-based and local-level segregation increased, but global modularity decreased (Fig. 5e; Table 2). Coinciding with the largest correlation, small-worldness was the largest predictor factor for identifying age ($\lambda = 0.35$; Fig. 5e; Table 2). Together, the results highlight a complex pattern of topological change from nine to 32 that can be characterized by increasing integration alongside decreasing global segregation and increasing local-level segregation. During this period, increasing small-worldness is particularly distinct, indicating that the network is becoming both more globally efficient and more locally specialized with age.

The second epoch of life ends around 32 years old (Fig. 4c; identified 97 times). At this age, there are many changes in the directionality of topological development. Before 32 years old, global efficiency increased while characteristic path length, small-worldness, modularity, and betweenness centrality significantly decreased. These correlations shift to the opposite direction after 32 years old (Table 2; Supplementary Fig. 7a). This result suggests a shift around 32 years old from increasing to decreasing integration as well as changes from decreasing to increasing modularity and betweenness centrality. In addition, the topological metric driving the relationship with age changes from small-worldness to local efficiency around 32 years old. Thus, the beginning of the fourth decade of life marks the end of a phase of increasing efficiency and integration and the start of a period of increasing segregation.

### Epoch 3: 32–66 years old "Adulthood"
The third epoch occurs from 32–66 years old ($n = 1092$), extending across three decades of adulthood. Across this period, 10 topological measures were significantly correlated with age, characterized by decreasing integration, general increases in segregation, and minimal centrality changes (Fig. 5d; Table 2). While clustering coefficient was most highly correlated with age, the LASSO retained local efficiency as the largest predictor of age across this period ($\lambda = 0.63$; Fig. 5d; Table 2). Importantly, clustering coefficient and local efficiency are highly correlated ($r = 0.91$; Supplementary Fig. 5a). Thus, both analyses indicate that the most notable feature in topology across this age range is increasing connectivity between neighboring regions. Clustering coefficient was significant in 71 regions and local efficiency was significant in 74 regions (after FDR correction), indicating this age-topology relationship was disrupted across the majority of the brain (Supplementary Fig. 6a, c). Together, these results suggest network integration decreased with minimal centrality changes, and while segregation was complex, most segregation metrics increased across this epoch.

The third epoch ends around 66 years old, which was the least distinct of the four major turning points (identified 44 times; Fig. 4c). While there are no significant changes in the directionality of topology at this age (Supplementary Fig. 7a), we observed the driving topological metric shift from local efficiency to modularity (Fig. 5a). Thus, this turning point reflects a shift in the topological features most predictive of age from increasing connectivity between neighboring nodes to increasing separability into highly interconnected groups.

### Epoch 4: 66–83 years old "Early aging"
The fourth epoch ranges from 66 to 83 years old ($n = 406$), spanning the shift from adulthood into early aging. Only four topological metrics showed significant correlation with age (Fig. 5c; Table 2). While this period is topologically most distinct in modular changes, decreasing integration and increasing centrality are also present

(Fig. 5c; Table 2). The regularization of the LASSO had to be weakened for any predictors to survive (see "Methods"; "Statistics"). The LASSO indicated that modularity is the strongest predictor of age, aligning with the largest correlation in this period ($\lambda = 0.20$; Fig. 5c; Table 2). The last turning point in the lifespan occurs around 83 years old, which was the second most frequently occurring turning point (Fig. 4c; identified 111 times). There were no significant changes in directionality at this age (Supplementary Fig. 7a); however, the most important factor for identifying age shifts from modularity to subgraph centrality. In other words, from this turning point onwards, the most age-associated topological change that occurs is that individual nodes in the network have increasing importance in local-level connectivity.

### Epoch 5: 83–90 years old "Late aging"

The last epoch is 83–90 years old ($n = 93$), which ranges from late aging individuals to the maximum age included in this study. Only subgraph centrality was significantly associated with age during their period (Fig. 5b; Table 2). Importantly, compared to all epochs, this epoch has the lowest statistical power due to sample size (mean power for epoch one = 0.72, epoch two = 0.97, epoch three = 0.92, epoch four = 0.35 and epoch five = 0.16). In addition, the regularization of the LASSO had to be weakened for any predictors to survive (see "Methods"; "Statistics"). With the less-sparse model, subgraph centrality was the strongest predictor of age, which aligns with the only significant correlation ($\lambda = 0.11$; Fig. 5b; Table 2). Importantly, subgraph centrality was only significantly correlated with age in 10 regions (Supplementary Fig. 6e), including the cuneus (right and left), the superior (right) and middle (left) occipital gyri, and the postcentral gyrus (right). Thus, an increase in centrality in late life has a spatial-temporal pattern. Together, these results suggest a potential reduction in the age-topology relationship, with increasing subgraph centrality as the most notable topological change of this period.

### Characterizing all turning points

Beyond detailing changes in topology *within* each epoch, it is helpful to compare topology differences *across* epochs. While UMAP provides information about where major turning points occur, we cannot interpret what is topologically changing at these points due to UMAPs having arbitrary dimensions (e.g., no loading scores). Simply put, UMAP informs us *where* non-linear changes occur, but not *what* those changes are. To explore *what* topological changes occur around these major turning points, we ran a Principal Components Analysis (PCA) with the 11 topological metrics across the entire lifespan, using a parallel analysis to identify three principal components (PCs) that explain 76.61% of the variance in topological measures (Fig. 6a; Supplementary Fig. 4b–e). While the fact that PCA is a linear method makes it less suitable for identifying where fluctuations occur in the data compared to manifold learning techniques, it is a useful tool for comparing the pre-defined epochs as it generates interpretable loading scores. Segregation measures load most heavily onto PC 1, while integration metrics load mostly on PC 2, and both segregation and centrality metrics load onto PC 3 (Fig. 6a; Supplementary Fig. 4b–e). PCA scores across epochs had significantly different variances and means in all PCs (Fig. 6a, c; Table 3).

We compared average PCA scores between consecutive epochs. Significant shifts in PC 1 and PC 2 occur between epochs one and two (PC 1 $p = 0.002$; PC 2 $p = 2.15 \times 10^{-10}$) and between epochs two and three (PC 1 $p = 4.95 \times 10^{-13}$; PC 2 $p = 0.002$) (Fig. 6a, c; Supplementary Fig. 7b). Neither epoch comparison had significant differences in PC 3 (epochs one and two $p = 0.692$; epochs two and three $p = 0.972$) (Supplementary Fig. 7b). These results suggest that the first two turning points – nine and 32 years old – identify significant shifts occurring in the two primary components, upon which load most segregation and integration metrics (Fig. 6a; Supplementary Fig. 4b, c). Epochs three and four—the 66-year-old turning point—is the only point where a

significant shift in PCA scores occurs across all PCs (Fig. 6a, c; Supplementary Fig. 7b; PC 1: $p = 1.01 \times 10^{-14}$; PC 2: $p = 2.47 \times 10^{-13}$; PC 3: $p = 1.82 \times 10^{-13}$). These results suggest a distinct shift across all primary components despite no directional changes in topology. The last turning point (83 years old) captures significant changes only in PC 2 ($p = 0.008$) but not PC 1 ($p = 0.912$) or PC 3 ($p = 0.065$) (Fig. 6a, c; Supplementary Fig. 7b). Together, these results indicate that differences in topology before and after 83 years old appear to be mostly within integration metrics, which load onto PC 2 (Fig. 6a; Supplementary Fig. 4c).

Lastly, we used the trajectories of PCA scores within epochs to examine differences in developmental patterns. Using dynamic time warping, we qualitatively compared the trajectory patterns between each consecutive epoch (Supplementary Fig. 7c) (see "Methods"; "Dynamic time warping"). The warping distance (Euclidean) conveys how different two trajectories are—larger distances indicate more different trajectory patterns than shorter distances. This analysis showed that epochs one and two have the most similar trajectory pattern, followed by epochs four and five and epochs two and three (Fig. 6b, c). The two epochs with the largest difference in trajectories are three and four, suggesting that the actual pattern of topological change is the most dissimilar before and after 66 years old, compared to any other turning point (Fig. 6b, c).

When comparing all analyses across all turning points, 32 years old emerges at the largest turning point across the lifespan (Fig. 6c). The last turning point—83 years old—appears to be the smallest (Fig. 6c). The two 'middle' turning points—66 and nine years old—are distinct from each other in that significant directionality changes occur around nine years old but none at 66 years old (Fig. 6c). Together, these results indicate that the major lifespan turning points signify critical shifts in the trajectory of topological development.

## Discussion

Our results emphasize the complex, non-linear topological changes that occur across the lifespan, with oscillating network integration development between childhood, adolescence, and adulthood. We found that centrality is important during adolescence but minimally for the rest of the life. Additionally, our results show a pattern of increased network segregation but a decline of the age-topology relationship in late life. Broadly, the trajectory of topological development can be distinctly separated into multiple phases of development, with four major turning points occurring around nine, 32, 66, and 83 years old. These points indicate where the trajectory of topological development shifts significantly and begins a new projection into a different area of the manifold space. To the best of our knowledge, manifolds have not previously been used to identify topological turning points, therefore we aim to review where these turning points align with important anatomical and contextual milestones.

The first turning point indicates that the childhood topological trajectory ends around nine years old. The first few years of life are marked by consolidation and competitive elimination of synapses[20] and rapid increases in gray and white matter volume[1]. Our results indicated that, topologically, structural networks develop along the same dimensions from birth until about nine years old. This is consistent with a previously identified cortical turning point around seven years old, where global efficiency reaches a minimum, cortical thickness peaks, and cortical folding stabilizes[49]. Thus, within the first decade of life, while myelination and white matter volume increase rapidly, topological efficiency decreases parallel to synaptic elimination. This age also aligns with the onset of puberty, which begins between eight and 13 years old for females and nine and 14 years old for males[50], marking the initiation of significant alterations in hormone expression[51] and robust neurological changes[52–55]. Coinciding with this topological and neurobiological shift, the transition from childhood to adolescence brings with it increased risk of mental health disorders[56],

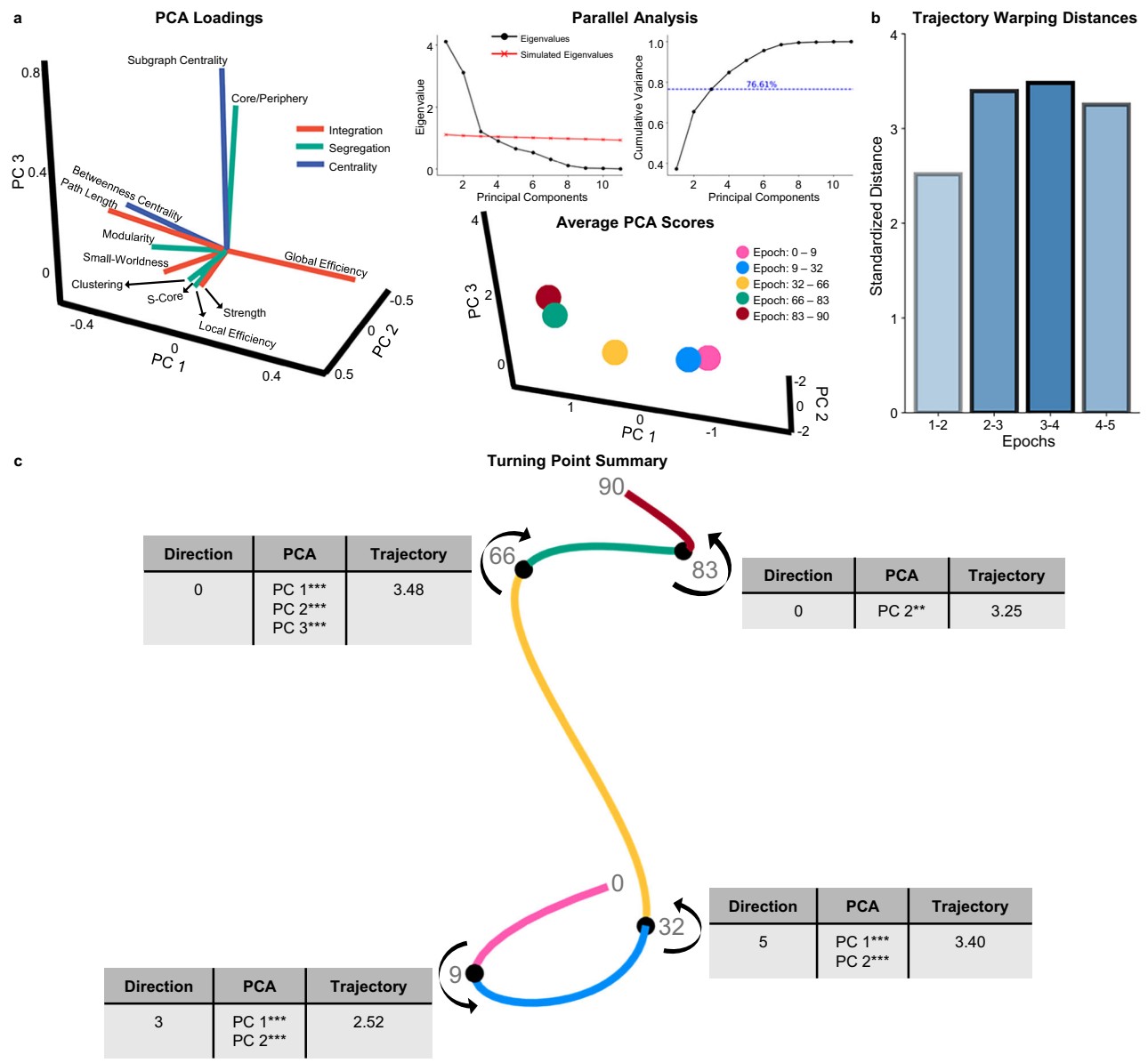

**Fig. 6 | Characterization of turning points using Principal Components Analysis (PCA).** **a** Parallel analysis shows that three PCs (principal components) should be retained. The three PCs account for 76.61% of the variance in topological data across the lifespan. On the left, there is a three-dimensional loading plot for the PCA. PC 1's four largest loadings are clustering coefficient, local efficiency, strength, and s-core. PC 2's four largest loadings are global efficiency, characteristic path length, betweenness centrality, and modularity. PC 3's four largest loadings are subgraph centrality, core/periphery structure, small-worldness, and modularity. On the right, average PCA scores for each epoch are plotted in 3-dimensional space.

**b** From the DTW analysis, the summed standardized warping distances across all PCs for consecutive epochs are plotted in a bar graph. **c** A turning point summary schematic shows a 3-dimensional manifold with turning points (black spheres) and demonstrates the direction of the projection of each epoch. Next to each turning point is a table with the total number of significant changes in Pearson correlations of individual topological metrics ('Direction'), significant shifts across PCs ('PCA') (assessed via Welch's ANOVA), and the total standardized warping distance ('Trajectory'). Together, this schematic characterizes changes that occur at each major turning point. *** indicates $p < 0.001$, ** indicates $p < 0.01$, * indicates $p < 0.05$.

progression in cognitive capacity[57], and modifications of socio-emotional and behavioral development[58,59]. Thus, the nine-year-old turning point not only signifies a distinct shift in topological development but also aligns with key cognitive, behavioral, and mental health developmental milestones.

The second lifespan epoch, ages nine to 32, indicates that the trajectory of topological development remains consistent across this period. While adolescence begins with puberty, the end of adolescence is less clear, with older definitions ending before 20 and more recent definitions extending into the mid-20s[60]. The transition to adulthood is influenced by cultural, historical, and social factors, making it context-dependent rather than a purely biological shift[61,62]. Our findings suggest that in Western countries (i.e., the United Kingdom and United

States of America), adolescent topological development extends to around 32 years old, before brain networks begin a new trajectory of topological development.

Additionally, 32 years old is the strongest topological turning point of the lifespan. At this age, the most directional changes and a large shift in trajectory occur compared to the other turning points. These findings are highly consistent with previous work exploring individual topological metrics[13–15] that identify significant peak/inflection points at the beginning of the fourth decade. Beyond organizational changes, this turning point aligns with developmental trajectories of white matter. White matter volume and fractional anisotropy peak around 29 years old[1,63,64], mean diffusivity arrives at a minimum around 36 years old[63,64], and radial diffusivity reaches a

**Table 3 | Statistics for comparing PCA scores across epochs**

| | | Principal Component Analysis | | | | |
|---|---|---|---|---|---|---|
| | | Epoch 1 | Epoch 2 | Epoch 3 | Epoch 4 | Epoch 5 |
| **PC 1** | **Variance** | 3.04 | 3.13 | 4.07 | 3.60 | 3.19 |
| | *Levene's Test* | $F = 15.06$, $p = 3.20 \times 10^{-12}$ | | | | |
| | **Mean (± STD)** | −0.78 ± 1.74 | −0.50 ± 1.77 | 0.62 ± 2.02 | 1.67 ± 1.90 | 1.85 ± 1.80 |
| | *Welch's ANOVA* | $F_{(573.90)} = 201.84$, $p = 5.91 \times 10^{-108}$ | | | | |
| **PC 2** | **Variance** | 2.57 | 3.08 | 2.89 | 3.72 | 3.39 |
| | *Levene's Test* | $F = 4.83$, $p = 6.97 \times 10^{-04}$ | | | | |
| | **Mean (± STD)** | −0.38 ± 1.60 | 0.10 ± 1.76 | 0.36 ± 1.70 | −0.48 ± 1.93 | −1.20 ± 1.85 |
| | *Welch's ANOVA* | $F_{(571.57)} = 38.79$, $p = 9.58 \times 10^{-29}$ | | | | |
| **PC 3** | **Variance** | 0.93 | 1.08 | 1.15 | 1.91 | 1.44 |
| | *Levene's Test* | $F = 9.25$, $p = 1.95 \times 10^{-07}$ | | | | |
| | **Mean (± STD)** | −0.15 ± 0.96 | −0.09 ± 1.04 | −0.07 ± 1.07 | 0.52 ± 1.38 | 0.90 ± 1.20 |
| | *Welch's ANOVA* | $F_{(566.14)} = 35.14$, $p = 3.13 \times 10^{-26}$ | | | | |

PC are principal components. STD is the standard deviation. Welch's ANOVA was one-way.

minimum around 31 years old[63]. Thus, across the phase of life, while white matter integrity and volume are increasing rapidly, topological structure at the macroscale is becoming more efficient and less segregated. Together, these results indicate that significant changes in white matter integrity and topological development occur around the beginning of the fourth decade of life.

After age 32, the longest epoch begins, covering three decades of adulthood until age 66. Compared to rapid maturation in earlier life, changes in network architecture slow during this period[39,63,64], which is consistent with our results that there are no major topological turning points until the 60s. Aligned with the slowing of white matter maturation during this period, the patterns of topological change are less complex than previous epochs, with clear increases in segregation and decreases in efficiency. This period of network stability also corresponds with a plateau in intelligence and personality[39]. Consequently, not only do we observe the alignment of turning points with significant anatomical and cognitive milestones, but also the stable topological epochs of life coincide with periods of anatomical, cognitive, and behavior consistency.

The third turning point, age 66, marks a topological shift without directional changes. Consistent with past work[13–15], we find no directional changes in network organization occurring at this age. However, there were significant differences in PCA scores in all PCs. Therefore, this turning point may reflect protracted or accelerated development. Indeed, accelerated decreases in white matter integrity are known to occur in late life[63]. This decrease in white matter integrity is generally referred to as 'age-related' degeneration, meaning reductions in white matter coherence are expected in late aging individuals[3,65]. Topologically, we find that during this phase macroscale reorganizational patterns simplifies—with the most distinct change being increasing modularity[13,66,67]. Together, these patterns suggest a sparsification of the structural network in aging. Additionally, the early 60 s mark an important shift in health and cognition in high-income countries, such as the onset of dementia and hypertension[68,69]. Hypertension, characterized by chronically elevated blood pressure, is linked to cognitive decline and accelerated brain aging and is also a known risk factor for dementia[70,71]. Thus, as with the first two turning points, age 66 also aligns with significant shifts in health and cognition.

The last turning point marks a distinct decline in the age topology. After 83 years old, only subgraph centrality retained a significant relationship with age. It is possible that the lack of significant findings reflects the small sample size ($n = 93$), which is reflected in the low statistical power in this epoch. However, when considering the significant correlations from previous epochs, a declining trend appears after middle age; epoch three had 11 significant correlations, epoch four had four significant correlations, and epoch five had one significant correlation. Therefore, this could reflect a true weakening relationship between age and structural brain topology in late life.

The data processing pipeline and manifold construction involve numerous design choices, and while we have attempted to test how these may impact our results, some caveats remain. First, we used four versions of the AAL90 atlas, warped to two-year, one-year, and neonatal brain sizes, to address early-life brain volume change[1,72,73]. This step was crucial for a consistent parcellation necessary for unbiased topological analysis, but atlas alignment differences may exist. Second, we harmonized tracked networks and provided 10 additional analyses exploring various harmonization methods (Supplementary Fig. 7). We chose the approach with the fewest remaining dataset effects. Notably, no turning points coincided with dataset transitions (e.g., BCP ends at five and CALM starts at six), as we would expect if turning points were dataset effects. However, harmonization may have over- or under-corrected for dataset differences. Third, networks were thresholded to a fixed density to ensure unbiased topological analysis, though this may obscure individual differences and small age-related changes. Additional analyses to assess the effects of these choices (Supplementary Fig. 1b) and variable density analysis demonstrate relatively consistent turning points (Supplementary Fig. 8d). Despite this consistency, density-controlled results must be interpreted in the context of thresholding. Third, we performed sensitivity analyses on turning point identification, which show generally consistent results, though it is important to note that the degree of the polynomial fit influences turning points (Supplementary Fig. 8). Additionally, our manifold spaces were constrained to three dimensions for straightforward interpretation; however, higher-dimensional UMAP embeddings could potentially reveal finer-grained but important turning points in topology. Finally, although we employed UMAP for its speed and its ability to capture both global and local structure in the data[35], applying other non-linear techniques—such as diffusion map embedding or t-SNE—could offer valuable context for assessing the robustness of these results.

Additional key limitations are present in the project design. Despite sex effects in individual organizational measures, we did not sex-stratify this data due to sample size considerations. Future work should explore if the four major turning points identified here are sex sensitive. Moreover, the cross-sectional design of this project, due to the limited availability of longitudinal lifespan datasets, limits exploration of causality or temporal dynamics within an individual. Additionally, while all participants included were deemed healthy by

respective project guidelines, the gap between a healthy older individual and their peers may be larger than that between a healthy middle-aged individual and their peers. It is reasonable to speculate that older individuals in this study are healthier than typical individuals their age, which could bias the older sample.

In conclusion, our findings suggest that structural topological development occurs non-linearly across the lifespan, with major turning points occurring around nine, 32, 66, and 83 years old. These ages demarcate periods of complex topological development with distinct age-related changes. This work reinforces the need for multivariate, lifespan, population-level approaches to deepen our understanding of complex topological development.

## Methods

### Datasets and preprocessing
This study includes nine separate datasets that were collected and preprocessed specifically to suit the age range for the sample. Details on dataset samples, imaging procedures and preprocessing are summarized in Supplementary Materials (Supplementary Table 1). Four datasets were preprocessed in-house using QSIprep[74], while five datasets were preprocessed by Dr Yeh and made publicly available on DSI studio's Fiber Data Hub (https://brain.labsolver.org/)[75] (Supplementary Table 1).

**In-house processed datasets.** The Baby Connectome Project (BCP) is a multi-site study conducted at the University of North Carolina at Chapel Hill and the University of Minnesota aimed at capturing the typical development of infants[76]. This dataset works as an extension of previous human connectome projects but is optimized for imaging and processing suitable for zero to five-year-olds[76] (Supplementary Table 1). During harmonization, we utilized all scans from infants 12 months or older; however, for the analysis, we excluded longitudinal and repeat scans by using only the first scan for every infant (Supplementary Table 1). Some individuals had two different types of scans within the same session – 6-shell or dir-79 (Supplementary Table 1). Due to previous reports that the 6-shell scheme resulted in increased accuracy of local fiber orientation estimates[76], if both scan types were available, the 6-shell scan was used.

The Centre for Attention, Learning and Memory (CALM), Resilience in Education and Development (RED), and Attention and Cognition in Education (ACE) datasets were collected at the MRC Cognition and Brain Sciences Unit at the University of Cambridge. The CALM cohort is a specialized sample of children who are neurodivergent[77]. All scans were included during harmonization; however, only neurotypical controls were included in the analysis (Supplementary Table 1). The RED dataset was aimed to sample children from diverse socio-economic (SES) backgrounds[7]. One participant was removed due to missing age data (Supplementary Table 1; resulting sample size of $n = 75$). The ACE dataset aimed to capture a realistic representation of SES across the UK[78] (Supplementary Table 1).

**DSI Studio semi-processed datasets.** Dr Yeh has preprocessed and made available many datasets on DSI Studio's Fiber Data Hub (https://brain.labsolver.org/)[75]. The dataset-specific preprocessing methods below are also published on the DSI Studio website.

The Developing Human Connectome Project (dHCP) is a collaborative effort between King's College London, Imperial College London, and Oxford University that collects neuroimaging data from neonates[79]. All longitudinal scans and infants born earlier than 37 weeks' gestation (preterm) were excluded from this analysis, resulting in a cross-sectional, term infant sample (Supplementary Table 1). The images were denoised and corrected for Gibbs ringing, motion, eddy current, and susceptibility artifact using the diffusion SHARD pipeline[80]. A quality check was conducted using neighboring

DWI correction (NDC)[81]. 34 scans (including repeated scans) were excluded due to their low NDC values identified by a median value-based outlier detector.

The Human Connectome Project Development (HCPd) aims to capture a diverse but typical developmental sample[82]. This multi-site study includes Harvard University, University of California-Los Angeles, University of Minnesota, and Washington University in St. Louis[82]. Sample and imaging information can be found in Supplementary Table 1 and in further detail Somerville et al.[82]. The susceptibility and eddy current artifacts were corrected using FSL topup and eddy (FMRIB, Oxford). The correction was conducted through the integrated interface in DSI Studio's ("Chen" release). The diffusion MRI data were rotated to align with the AC-PC line. The accuracy of b-table orientation was examined by comparing fiber orientations with those of a population-averaged template[83].

The Human Connectome Project Young Adult (HCPya) is a multi-site study collected by the Washington University-University of Minnesota Consortium of the Human Connectome Project (WU-Minn HCP), which aims to capture a large sample of healthy adults[84]. Sample and imaging information can be found in Supplementary Table 1 and in further detail Van Essen et al.[84]. A group average template was constructed from a total of 930 subjects. The diffusion data were reconstructed in the MNI space using q-space diffeomorphic reconstruction[85] to obtain the spin distribution function[36].

The Human Connectome Project Ageing (HCPa) is a multi-site study aimed at capturing healthy aging from 36 to 100+ years old[86]. The sample used in this analysis excluded participants scanned at 100+ years old ($n = 12$), resulting in a cross-sectional sample ranging from 36 to 90 years old (Supplementary Table 1; $n = 706$). Further details on the HCPa sample and imaging methods can be found at Bookheimer et al.[86]. The susceptibility and eddy current artifacts were corrected using FSL topup and eddy (FMRIB, Oxford). The correction was conducted through the integrated interface in DSI Studio's ("Chen" release). The diffusion MRI data were rotated to align with the AC-PC line.

The Cambridge Centre for Ageing and Neuroscience (camCAN) project aims to capture age-related changes in neurocognitive systems[87]. This project is conducted at the MRC Cognition and Brain Sciences Unit, University of Cambridge, and focuses on exploring important aspects of health in aging. Sample and imaging information can be found in Supplementary Table 1 and in further detail Shafto et al.[87]. The b-table was checked by an automatic quality control routine to ensure its accuracy[88].

For dHCP, HCPd, and HCPa, the accuracy of b-table orientation was examined by comparing fiber orientations with those of a population-averaged template[81]. The restricted diffusion was quantified using restricted diffusion imaging[89]. Additionally, with dHCP, HCPd, HCPa and camCAN, the diffusion data were reconstructed using generalized q-sampling imaging[36] with a diffusion sampling length ratio of 1.25. Alternatively, for HCPya, a diffusion sampling length ratio of 2.5 was used, and the output resolution was 1 mm.

### Connectome construction
**Tractography.** All networks were tracked using standard GQI plus deterministic tractography in DSI Studio[36]. For participants three years old and older, the QSIprep dsi_studio_gqi workflow[74] was applied with the AAL116 atlas[90]. All other participants were tracked directly in DSI Studio using multiple versions of the AAL atlas. Participants aged 24 to 35 months were tracked with the AAL90 two-year-old atlas[73], those aged 12 to 23 months with the AAL90 one-year-old atlas[73], and those younger than 12 months with the AAL90 neonatal atlas[73]. For all networks parcellated with the AAL116 atlas, we removed additional subcortical regions (numbers 91–116), which resulted in the AAL90 atlas. This progressive use of AAL90 atlases with the same regions fit to different brain volumes enables direct comparison between regions

across the lifespan while accommodating for drastic brain growth in the first two years of life. All tracking was performed with the same parameters—maximum fiber length of 250 mm, minimum fiber length of 30 mm, 5 million streamlines, random seeding, 1 mm step size, and turning angle 35°. We used count-end connectivity, indicating that streamlines were identified between two regions if the streamline ended in both regions.

**Harmonization.** All data, including longitudinal and repeat scans in BCP and neurodivergent group in CALM, were included in harmonization ($N = 4216$). Multiple harmonization methods for variable density and density-controlled networks as well as assessing efficacy of harmonization before and after thresholding and can be found in the supplement (Supplementary Fig. 8). The harmonization methods were evaluated by the total number of FDR-corrected significant effects of study within age-bins across density, modularity, core/periphery structure, global efficiency, average degree, and average strength (see "Methods"; "Graph Theory"), as well as visual inspection of generalized additive models. We determined that our 'double harmonized' method before thresholding was the most effective across both variable and density-controlled analyses. Double harmonization was performed using ComBat[37] to harmonize across atlas and then harmonize again across study (Fig. 1c). For each step, a mask was used to only retain connections that were present before harmonization in addition to setting any negative connections produced by harmonization to zero. Covariates that were preserved during harmonization included participant ID, age, sex, and neurodiversity group to identify children in CALM who are neurodiverse.

**Thresholding.** Before thresholding, 14 participants were identified as outliers (dHCP $n = 1$; CALM $n = 2$; RED $n = 1$; ACE $n = 1$; HCPya $n = 3$; HCPa $n = 1$; camCAN $n = 5$) due to having network density above or below three standard deviations for the age bin (comprised of the closest rounded year) and were removed. With this reduced sample ($n = 4,202$), two thresholding methods were performed – (1) preserve variable density across the lifespan and (2) control density across the lifespan for topological comparison.

For the variable density analysis, we performed a generalized additive model (see "Methods", "Statistics") on the raw network densities and took 70% of the regression to obtain a 'target' density for each age (Supplementary Fig. 1a). Then, for each age group within each study, we applied the absolute threshold based on streamline count cut-off that yielded an average density equal to the target density for that age. The resulting networks were thresholded to densities ranging from 21 to 8%, with the original relationship between age and density preserved (Supplementary Fig. 1a). These networks were then used only in the connectivity analysis to explore density, degree, and strength of networks.

Additionally, the density-controlled networks were constructed for topological analyses. These networks were thresholded by the streamline counts so that each individual, regardless of age, had a 10% dense network. 10% was utilized because the sparsest network in the sample was 11%. Thus, 10% was the highest possible density where every network in the sample is thresholded, as well as being consistent with past lifespan work[91]. Additional densities of 8% and 5% can be found in the supplement (Supplementary Fig. 1b). All networks were converted to normalized weighted networks using weight_conversion() from the Brain Connectivity Toolbox[38], which rescales all weights to range from 0 to 1.

## Graph theory
All graph theory metrics were calculated using the Brain Connectivity Toolbox (BCT) in MATLAB 2020b[38]. Global measures included network density, modularity, global efficiency, characteristic path length, core/periphery structure, small-worldness, k-core, and s-core, while local measures utilized were degree, strength, local efficiency, clustering coefficient, betweenness centrality, and subgraph centrality. All local measures were averaged across the network for the topological analysis, though local-level correlations between measures and age are in the supplement (Supplementary Fig. 6). Modularity was calculated at one spatial resolution (gamma = 0.6), which was chosen after sweeping through gamma values from 0.2 to 2. At each level, observed modularity was compared to the modularity of randomized networks with preserved density and degree distributions. The spatial resolution was decided based on which level had the largest Kolmogorov-Smirnov (KS) statistic[92], indicating that the modularity structure was the most non-random (Supplementary Fig. 9). Summaries of all measures can be found in the supplement (Supplementary Table 2). All topological measures were assessed using generalized additive models in RStudio version 4.1.2[93]. In these models, cubic regression splines were used to smooth across age, and sex, atlas, and dataset were controlled for.

## Uniform Manifold Approximation and Projection (UMAP)
To project topological data into a manifold space, we used the UMAP package in Python version 3.7.3[35]. Before data was put into the UMAP, it was first standardized using Sklearn's StandardScalar()[94]. UMAP requires four pre-defined parameters—minimum distance and nearest neighbors, number of components, and distance metric. Minimum distance typically ranges between zero to one and determines how closely data points are packed together in the low-dimensional representation (low values result in more clustered representations)[35]. Nearest neighbors defines the size of local neighborhoods when learning the manifold structure[35]. This parameter, therefore, determines the balance between local versus global structure—a low nearest neighbors value pushes the UMAP to capture more local structure and vice versa. Nearest neighbors can be at minimum two or at maximum one less than the length of the data input. The number of components simply determines how many dimensions the projection should be embedded in. We predefined this as three dimensions in an effort to capture multi-dimensional changes without losing interpretability. Lastly, the distance metric determines how the distance is calculated. We used the Euclidean distance.

A limitation of UMAP is that the minimum distance and nearest neighbors parameter choice greatly determines the shape of the projected manifold[48]. While UMAP always captures patterns within the data, the parameter choices alter which patterns are projected, making it challenging to derive meaningful interpretations of the projections. To mitigate this, we derived 968 combinations of UMAP parameters. The nearest neighbor parameter was set to 88 whole numbers that ranged from two to 89, while the minimum distance parameter was 11 values evenly spaced, ranging from 0.1 to one. Thus, we conducted our analysis on a complete range of UMAP projections, from manifolds representing mostly local patterns through manifolds capturing mainly global patterns.

## Turning point identification
To determine what constitutes a turning point, we have constructed our own algorithm with multiple parameters. First, we must find a line of best fit through the 3-dimensional manifold. In Python version 3.7.3, we created three polynomial fits—one for each dimension—which requires the choice of the polynomial degree (Fig. 4a). The equation for each dimension is as follows:

$$Dimension(age) = \beta_0 + \beta_1 age + \beta_2 age^2 + \beta_3 age^3 + \beta_4 age^4 + \beta_5 age^5 + \epsilon$$

$$(1)$$

Polynomials were fit using the polyfit() function from the *numpy* package, which uses least squares error[95]. Together, these polynomials

create the 3D line of best fit through the manifold space. For our main analysis, we fit 5-degree polynomials and have included iterative polynomials ranging from two to 12 in the supplement (Supplementary Fig. 10a). This sensitivity analysis highlights that a degree of five is a middle-ground between visually underfit and overfit lines, with high-degree lines including more middle-age turning points (e.g., between 50-70 years old). Importantly, turning points occurring around 10, 30, and 80 are robust across most degree choices (Supplementary Fig. 10a). Generally, the choice of degree impacts *where* in the lifespan turning points are identified.

We then calculated the gradients of the lines of best fit and identified points where the gradient changes sign (positive to negative or negative to positive) along each dimension. Small fluctuations were then filtered to remove minor inflections by removing points where the sum of the gradients around the point was relatively small. This filtering process requires two parameters – a gradient window (W) and a gradient threshold (T). The gradient window determines the number of years around the inflection point ($i$) that will be the scope of the gradient threshold. The gradient threshold is the cut-off for how large the sum of the absolute value of gradients within this range needs to be to be retained. An inflection point will survive this cutoff if:

$$T < \sum_{i-W}^{i+W} |G_i| \qquad (2)$$

$G_i$ represents the gradient at $i$ year, W is the gradient window, and T is the gradient threshold. The larger the gradient threshold, the sharper the inflection point (i.e. steep slopes on either side of the point) must be in order to be kept in the analysis. For our analysis, we defined the gradient window to five years and the gradient threshold to 0.8, though it is important to note that many variations of these parameters result in the same turning points being identified. Sensitivity analysis of varying gradient thresholds can be found in the supplement (Supplementary Fig. 10c). This analysis shows that turning points are stable across gradient thresholds 0.1 to 0.9. Turning points around eight and 83 are retained at a gradient threshold of 1.2. This indicates that the first and last turning points of the lifespan are the 'largest' or 'sharpest' in terms of the change in slope through the manifold space (Supplementary Fig. 10c). Thus, this parameter affects the sensitivity to the *size* of turning points but not where the turning points are located across the lifespan.

The second step in identifying turning points in a manifold is to handle instances where multiple points have been detected in close proximity. For example, if age 31 in dimension X and age 33 in dimension Y were identified as inflection points, we interpret these as representing a single turning point rather than two distinct trajectories, given their proximity. This process requires an age window parameter (A) to determine the age range around the inflection point in which a mean will be calculated. This averaging procedure occurs both within and across dimensions. Average turning points are then rounded and considered the 'final' turning points. For our analysis, we used an age window parameter of five years and have included a sensitivity analysis to explore how changing the age window affects the turning points identified which can be found in (Supplementary Fig. 10b). Between age windows of one through 10, we see no changes in turning points beyond a single year (Supplementary Fig. 10b). Thus, this parameter effects *where* a turning point is identified, similar to the degree of the polynomial, though its influence is minimal.

Turning points were identified for each of the 968 UMAP projections. *Major* turning points were defined by the peaks in the Gaussian kernel density function of all turning points (Fig. 4c). Thus, these points are the most frequent ages identified as turning points across all manifolds. We also assessed turning points in variable density networks (Supplementary Fig. 10d) and sex-stratified projections that have been mapped to the combined UMAP space using orthogonal

procrustes[96] (Supplementary Fig. 5). This analysis demonstrates that major turning points appear around similar ages for variable density networks and sex-stratified samples as those calculated in density-controlled networks, with some variation around the 66 turning point. Thus, our conservative thresholding for easy topological interpretability does not appear to drastically change where in the lifespan major turning points have been identified. Major turning points mark the average age at which topological data begins a new trajectory through the manifold, indicating a distinctly different organizational change across age. Thus, between major turning points, we define age epochs in which topological change is occurring along the same trajectory through the manifold space.

### Epoch correlations
We applied Pearson correlations within epochs (i.e., age ranges between major turning points) to explore changes in each organizational metric across age. We used correlations to examine *between* epochs simply by identifying when significant correlations in consecutive epochs changed direction (i.e., from a positive correlation to a negative correlation and vice versa) (Supplementary Fig. 7a). We also provided local-level correlations between measures and age within each brain region after false-discovery rate (FDR) correction[97] (Supplementary Fig. 6).

### Least Absolute Shrinkage and Selection Operator (LASSO)
To explore driving topological factors within epochs, we employed regularized LASSO models[98] in MATLAB 2020b with 10-fold cross-validation (CV) to perform variable selection with multicollinear predictors. The benefit of LASSO models is that they penalize the absolute value of coefficients, which results in some coefficients being pushed to zero, allowing for easy interpretation of important model features[98]. This penalization term is multiplied by a constant, λ, which is determined through the 10-fold CV. 10-fold CV trains the LASSO on nine folds (i.e., a subset of the data) and is tested on the 10th fold. To encourage sparsity in the model, we selected the largest lambda where the mean squared error (MSE) is within one standard error of the minimum MSE. For the two epochs, this level of sparsity resulted in no variables selected, and therefore, the LASSO model for epochs four and five was created by selecting the lambda value with the minimum MSE.

### Principal Components Analysis (PCA)
We conducted a PCA[99] in MATLAB2020b to reduce the dimensionality of graph theory metrics for the purpose of exploring between-epoch changes. After standardizing the data, we ran a PCA with the maximum number of components (11) and conducted a parallel analysis to determine how many components to retain[100]. For the parallel analysis, we created 1,000 iterations of standardized random data and conducted PCAs for each iteration. We then calculated the top 95% confidence interval of eigenvalues produced by the random samples[100]. Components from the original PCA with eigenvalues exceeding the threshold set by 95% confidence interval from the random eigenvalues were retained as this indicates that eigenvalues were larger than expected by chance. This analysis indicated three components to be retained (Fig. 6a). A second PCA was run, this time constrained to three components which convey 76.61% of variance across the sample (Fig. 6a; Supplementary Fig. 4). To improve loading interpretation, an orthogonal rotation was applied using rotatefactors() with the varimax method[101].

We compared epochs based on their PCA scores, first using Levene's Test for Relative Variation[102] in Python version 3.7.3 to determine if the variance of PCA scores significantly differed across epochs. Since this test was significant, we used Welch's Analysis of Variance (ANOVA)[103] in Python to assess significant differences in mean PCA scores across epochs. Lastly, we ran post hoc Games-Howell[104]

tests in Python to determine which consecutive groups were significantly different (Supplementary Fig. 7b). The full table of all Games-Howell comparison outcomes can be found in the supplement (Supplementary Table 5). In some instances, *p*-values were set equal to zero due to the truncated precision of Python. In these cases, *p*-values are reported as less than the minimum printable value, $p < 1.00 \times 10^{-323}$.

### Dynamic Time Warping (DTW)

We also examined differences between epochs using DTW on PCA score series conducted in Python version 3.7.3[105]. DTW warps two time series to their optimal alignment. The algorithm calculates the local Euclidean distances between points in each series, calculating the global alignment between the series as the warping path that minimizes the sum of distances between series[105]. DTW distance, defined as the minimum cumulative distance of the warp, quantifies how far points in one series must shift to align with another, providing insight into differences in their shapes. For our analysis, we constructed a series for each epoch across each PC, represented as the average PCA score for each age (Supplementary Fig. 7c). The DTW distances for optimal warping between consecutive epochs were standardized within each principal component. These distances were then qualitatively compared−with larger distances suggesting more disparity between the *shape* of those series' trajectories.

### Reporting summary

Further information on research design is available in the Nature Portfolio Reporting Summary linked to this article.

## Data availability

The derived data generated in this study are available at https://osf.io/7p4y3/. CALM raw data are available at https://portal.camide.cam.ac.uk/overview/1158. Human Connectome Project raw data are available at https://nda.nih.gov/. CamCAN raw data are available at http://www.mrc-cbu.cam.ac.uk/datasets/camcan/. The semi-processed data from dHCP, HCPd, HCPya, HCPa, and camCAN used in this publication are available at https://brain.labsolver.org/. Source data are provided with this paper.

## Code availability

All code is available at https://github.com/alexamousley/lifespan_topological_turning_points.

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

## Acknowledgements

Alexa Mousley is supported by the Gates Cambridge Foundation. Richard Bethlehem is supported by an Academy of Medical Sciences Springboard Award and HDR-UK. Duncan E. Astle is funded by the James S. McDonnell Foundation Opportunity Award and the Templeton World Charity Foundation, Inc. (funder DOI 501100011730) under the grant TWCF-2022-30510. All research in the Department of Psychiatry at the University of Cambridge is supported by the National Institute for Health and Care Research Cambridge Biomedical Research Centre (NIHR203312) and the NIHR Applied Research Collaboration East of England. All data provided on DSI Studio were analyzed using the US National Science Foundation, ACCESS program, resource allocation (TG-CIS200026) at Extreme Science and Engineering Discovery Environment (XSEDE) resources (Towns, J. et al. Computing in Science and Engineering 16, 62-74 2014) by Fang-Cheng Yeh. We want to thank the research teams at the MRC Cognition and Brain Sciences Unit that have collected RED, ACE, CALM, and camCAN datasets. The RED and ACE datasets were supported by the Templeton World Charity Foundation (TWCF0159) and by the UK Medical Research Council, UK (MC-A0606-5PQ41). CALM funding was provided by the UK Medical Research Council and the University of Cambridge, UK. Data used in the preparation of this work were obtained from CALM resource—https://calm.mrc-cbu.cam.ac.uk/. We would like to thank all members of the CALM Team for their help with recruitment, data collection, and data management, as well as all of the children and parents for their participation in the study. The CALM Team includes lead investigators Duncan Astle, Kate Baker, Susan Gathercole, Joni Holmes, Rogier Kievit, and Tom Manly. Data collection is assisted by a team of researchers and PhD students that includes Danyal Akarca, Joe Bathelt, Marc Bennett, Madalena Bettencourt, Giacomo Bignardi, Sarah Bishop, Erica Bottacin, Lara Bridge, Diandra Brkic, Annie Bryant, Sally Butterfield, Elizabeth Byrne, Gemma Crickmore, Edwin Dalmaijer, Fánchea Daly, Tina Emery, Laura Forde, Grace Franckel, Delia Furhmann, Andrew Gadie, Sara Gharooni, Jacalyn Guy, Erin Hawkins, Agnieszka Jaroslawska, Sara Joeghan, Amy Johnson, Jonathan Jones, Silvana Mareva, Elise Ng-Cordell, Sinead O'Brien, Cliodhna O'Leary, Joseph Rennie, Ivan Simpson-Kent, Roma Siugzdaite, Tess Smith, Stepheni Uh, Maria Vedechkina, Francesca Woolgar, Natalia Zdorovtsova, Mengya Zhang, and Alicja Monaghan. The authors wish to thank the many professionals working in children's services in the South-East and East of England for their support and to the children and their families for giving up their time to visit the clinic. We would also like to thank the radiographers who support the excellent pediatric scanning at the MRC CBU. The authors wish to thank the many professionals working in children's services in the South-East and East of England for their support, and to the children and their families for giving up their time to visit the clinic. We want to the Danyal Akarca for processing the RED, ACE and CALM datasets with QSIprep. Lastly, the camCAN project was funded by the UK Biotechnology and Biological Sciences Research Council (grant number BB/H008217/1), together with support from the UK Medical Research Council and University of Cambridge, UK. CamCAN data can be obtained from the camCAN repository (available at http://www.mrc-cbu.cam.ac.uk/datasets/camcan/). We also want to thank the extensive research teams and participants from the Human Connectome Projects. For the dHCP, the research leading to these data has received funding from the European Research Council under the European Union Seventh Framework Programme (FP/20072013)/ERC Grant Agreement no. 319456. The work was also supported by the NIHR Biomedical Research Centres at Guys and St Thomas NHS Trust. We are grateful to the families who generously supported this trial. We would like to acknowledge core support for data acquisition was provided by the Wellcome/EPSRC Centre for Medical Engineering [WT 203148/Z/16/Z]. We are also thankful to the WU-Minn-Oxford Human Connectome Project consortium (1U54MH091657-01) for access to their computing resources. For the BCP, we want to thank the members of the ELAB at the University of Minnesota who contribute to the BCP including Angela Fenoglio, Colleen Doyle, Elizabeth Sharer, Robin Sifre, Carolyn Lasch, Ella Coben, Sooyeon Sung, Kristen Gault, Rachel Roisum, Patrick Johnson, and Laura Thomas; Soma Prum, Amber Leinwand, and Jordan Jimenez at the University of North Carolina at Chapel

Hill; and we're indebted to the families who so generously contribute to our research. BCP is supported by the NIH grant (1U01MH110274) and the efforts of the UNC/UMN Baby Connectome Project Consortium, a K award (K01MH109773) to L. Wang, and a NIMH Biobehavioral Research Award for Innovative New Scientists (BRAINS) award (R01MH104324) to J. Elison. The HCPd is supported by the National Institute of Mental Health of the National Institutes of Health under Award Number U01MH109589 and by funds provided by the McDonnell Center for Systems Neuroscience at Washington University in St. Louis. The HCP-Development 2.0 Release data used in this report came from https://doi.org/10.15154/1520708. For HCPya, the data is provided by the Human Connectome Project, WU-Minn Consortium (Principal Investigators: David Van Essen and Kamil Ugurbil; 1U54MH091657) funded by the 16 NIH Institutes and Centers that support the NIH Blueprint for Neuroscience Research; and by the McDonnell Center for Systems Neuroscience at Washington University. Lastly, the HCPa is supported by the National Institute on Aging of the National Institutes of Health under Award Number U01AG052564 and by funds provided by the McDonnell Center for Systems Neuroscience at Washington University in St. Louis. The HCP-Aging 2.0 Release data used in this report came from https://doi.org/10.15154/1520707.

## Author contributions

A.M. performed fiber tracking, constructed connectomes, conducted the analysis, and drafted the manuscript. F.C.Y. preprocessed, quality controlled, and reconstructed the majority of networks. A.M., D.E.A., and R.B. conceptualized the analysis. D.E.A. and R.B. provided critical manuscript reviews and edits.

## Competing interests

A.M., D.E.A., and F.C.Y. declare no competing financial or non-financial interests. R.B. declares he is a co-founder of and holds equity in Centile Bioscience Inc.
