## [Transparent Peer Review file · Nature Communications]

Topological turning points across the human lifespan

Corresponding Author: Ms Alexa Mousley

Version 0:

Reviewer comments:

Reviewer #1

(Remarks to the Author)

Summary:

Mousley et al., describe trends of brain network topology across the human lifespan. Specifically, they used diffusion MRI data from several cohorts to construct structural brain networks spanning ages 0 to 90 years and then compute a range of graph-theoretical measures. Their findings reveal age-related trajectories in these metrics, and highlight four major turning points that delineate epochs of brain development and aging.

The scope of this paper is relevant to both network neuroscience and lifespan research. Furthermore, the authors' effort to integrate multiple datasets is commendable, and the analyses are generally rigorous and well presented. I am overall supportive of the work. However, there are a few points the authors should address to make the paper robust, accessible, and interpretable.

Major comments:

One of my initial observations is that the graph theoretical measures are introduced in a list-like manner, only broadly grouped into categories such as integration, segregation, and centrality indices. However, the manuscript would benefit from a more in-depth rationale for including each measure. First of all, what do integration, segregation, and centrality mean in the context of brain networks? Then, it would be helpful to briefly explain what each metric captures in terms of brain network organization and why it is important in the context of the lifespan. Maybe they have been associated with specific cognitive functions in the early lifespan, or are biomarkers of neurodegenerative disease? Providing this context would enhance the interpretability of the results and help readers understand the functional relevance of the observed trends.

One recommendation, then, would be to validate the results against null models. Specifically, the authors could generate random networks that preserve key properties—such as density and degree distribution—and compute the same graph-theoretical indices on these surrogate data. Comparing the observed trends to those derived from the null models would help establish whether the identified trends and turning points are statistically meaningful and not simply a byproduct of network size or structure. Incorporating null models would strengthen the robustness of the findings and support the interpretation that the reported age-related changes reflect biologically relevant patterns.

Regarding the more operational aspects of the projects, three points require clarification.

1. The method for pruning connections to correct for density is unclear. Specifically, it is not stated which connections were retained—were they the strongest connections or the most frequently observed across the population or age bins? The function `threshold_absolute` referenced in the GitHub repository appears to be missing, and the text does not explain it clearly. Both approaches are justifiable: retaining the strongest connections highlights those most reliably identified by tractography, whereas preserving the most common and maybe also weaker connections acknowledges their documented presence in the brain and their role in linking distant brain regions [1] and promoting inter-areal diversity in connectivity profiles [2]. Regardless of the chosen method, it should be clearly stated and justified in the text to enhance transparency and reproducibility.

2. The authors computed modularity using the function `modularity_und`, providing only the adjacency matrix as input. This choice effectively sets the resolution parameter to its default value ($\gamma=1$) and thus constrains the analysis to a single

spatial resolution. While this approach is sometimes used in the network neuroscience literature, it is well established that brain networks exhibit modular organization at multiple scales [3,4], often organized in a hierarchical manner [5]. Best practices recommend to identify modules and modularity by sweeping the γ parameter across a range of values to fully characterize modular organization. Therefore, the authors should clearly state that their results are based on a single partition and resolution, justify this choice, or present it as a limitation. A relevant reference for the paper in general and specifically in this context is [6], where age-related trends of modularity have been analyzed across the lifespan at a single scale, and the analysis was replicated at different spatial resolutions in the supplementary material. Additionally, the way modularity is described in the text is not technically accurate (line 110 page 5). Modularity is not defined as “the division into non-overlapping, highly-interconnected node groups”. Rather, it is a quality function that quantifies how well a network can be partitioned into such groups. This distinction is important for correctly interpreting the metric.

3. In the script `D_calculate_organization_measure.m`, the characteristic path length and global efficiency are computed based on a connection length matrix “`dist`”, which is a transformation of the structural connectivity matrix. However, the transformation is performed using the function `distance_bin.m`, instead of its weighted equivalent for undirected networks (`distance_wei.m`). Could the authors clarify this choice?

I found inconsistencies in the descriptions of age-related trends in the 5 epochs.

Epoch 1: 0-8 years old. The authors say that there is a decrease in small-worldness (line 200), whereas instead it increases in this age range.

Epoch 2: 8-32 years old. The authors first say that there is a decrease of network integration (lines 219,220) and then they say the opposite (line 225). I believe global integration increases in this epoch. Moreover, at line 235 they state “betweenness centrality happens around 32 years old”. What does it mean?

Minor comments:

It would be interesting (maybe as a supplemental analysis) to explore the regional patterns underlying the local graph-theoretical indices. While their average is used to determine turning points of the lifespan, a local analysis could reveal which specific brain areas are driving these trends within each of the five epochs. For example, what are the brain regions majorly driving the decline of betweenness centrality in the second epoch? Are these brain regions also driving the subsequent rise of that metric?

At line 85 (page 5) and 561 (page 23) the authors mention age bins. How are they defined? Do they comprise a single year of the lifespan or multiple?

At line 355 the authors say “centrality is important during adolescence but minimally for the rest of the life”. I think this sentence is imprecise. Centrality is always important, maybe the authors mean that variations in centrality measures show the most prominent age-related trends during adolescence?

At line 378 the authors say “the second lifespan epoch, ages 8 to 20, indicates no significant shift in the trajectory topological development”, but didn't they observe a shift in the trends of topological organization?

Lastly, the authors mention “extended data” (page 5 line 95), and “statistics” (page 23 line 565). However, at least to me, it is not clear where to find these sections. Are these files separate from the Supplementary Information?

References:

- [1] Distance-dependent consensus thresholds for generating group-representative structural brain networks, 2019, R.F. Betzel, Al. Griffa, P. Hagmann, B. Mišić, *Network Neuroscience*.
- [2] Specificity and robustness of long-distance connections in weighted, interareal connectomes, 2018, R.F. Betzel and D.S. Bassett, *PNAS*.
- [3] Multiscale Network Organization in the Human Brain, 2013, D. Bassett and F. Siebenhühner, in *Multiscale analysis and nonlinear dynamics*
- [4] Multiscale brain networks, 2017, R.F. Betzel and D.S. Bassett, *Neuroimage*
- [5] Hierarchical modularity in human brain functional networks, 2009, D. Meunier, R. Lambiotte, A. Fornito, K.D. Ersche, E.T. Bullmore, *Frontiers in Neuroinformatics*
- [6] The modular organization of brain cortical connectivity across the human lifespan, 2020, M.G. Puxeddu, J. Faskowitz, R.F. Betzel, M. Petti, L. Astolfi, O. Sporns, *NeuroImage*

(Remarks on code availability)

The results are mostly reproducible. There is one function that is missing (see comments for the authors)

Reviewer #2

(Remarks to the Author)

This paper investigates the non-linear development of structural brain topology across the human lifespan by analyzing diffusion imaging data from 4,216 participants aged 0–90 years. The authors compute 12 graph theory metrics to quantify network organization and then use Uniform Manifold Approximation and Projection to project these high-dimensional data into a lower-dimensional manifold space. By doing so, they identify four major topological “turning points” at approximately 8,

32, 62, and 85 years, which delineate five distinct epochs of brain network development. The study links these turning points to changes in integration, segregation, and centrality measures and discusses how these findings align with known neurobiological milestones and cognitive developmental phases. The study is novel and interesting, developing an innovative approach to mapping lifespan changes in brain network topology.

Specific comments

1. It is unclear whether the authors accounted for potential confounding variables, such as ethnicity, socioeconomic status, or other factors, in their analyses. Clarifying this would strengthen confidence in the generalizability of the reported findings.
2. The manuscript effectively employs UMAP to identify turning points in brain network topology across the lifespan. To further assess the robustness of these findings, please comment on the sensitivity of the identified turning points to the choice of dimensionality reduction technique. For instance, it would be informative to discuss whether linear methods (e.g., PCA, PLS), which provide interpretable component loadings, might yield comparable turning points. Similarly, a discussion comparing UMAP's results to those potentially obtained from other non-linear methods, such as diffusion map embedding, would be valuable. Additionally, please clarify the rationale for reducing the topological data specifically to three dimensions. Could exploring higher-dimensional UMAP embeddings potentially reveal finer-grained, yet meaningful, epochs of topological development beyond the five major ones reported?
3. The authors use polynomial fitting on age-averaged UMAP projections to identify turning points, and they have clearly made efforts to optimize the fitting parameters and smooth out small fluctuations. However, it is unclear why the same approach used to model the 12 topological metrics, namely, fitting GAMs to individual-level data, was not applied directly to the individual UMAP projections. Would this alternative approach yield similar turning points, and might it avoid potential information loss introduced by age-averaging? It would be helpful if the authors could justify their methodological choice and discuss whether the findings are robust across different modeling strategies.
4. This paper effectively delineates key topological transitions across the human lifespan. The authors employed three distinct methodological approaches (Pearson's correlation, Lasso regression, and Principal Component Analysis) to characterize the key metrics at each stage of structural development. However, the Discussion section would be strengthened by a more comprehensive interpretation of the neurobiological basis of these graph metrics. It is suggested that the authors expand the manuscript to consider how the key metrics identified within and between epochs may relate to specific white matter maturation or degeneration, and the contribution of these processes to the observed lifespan epochs.
5. In the Discussion, the authors interpret the decline in significant age–topology correlations across later life epochs as evidence of a weakening relationship between age and structural brain topology. However, could this pattern be confounded by reduced sample sizes in older age groups (e.g., from age 30s to 90s)? Clarifying whether statistical power was considered in this interpretation would strengthen the argument.
6. In Extended Data Figure 4b, the three boxplots appear identical, despite differing significance indicators. Could the authors double-check the accuracy of this figure and verify that all visualizations throughout the manuscript correctly reflect the underlying data?

(Remarks on code availability)

Version 1:

Reviewer comments:

Reviewer #1

(Remarks to the Author)

The authors have responded to all the issues raised in the initial review, and I appreciate the additional analyses they have carried out to address my comments. I find the revised manuscript suitable for publication.

There is one aspect where our perspectives do not fully converge: the choice to compute network distances on the binarized matrix. Based on the response, it appears that this decision was made a posteriori, after observing the dispersion of distance values from both weighted and unweighted graphs. In my view, this is not a fully convincing rationale. Ideally, such a methodological choice should be guided by a theoretical hypothesis—namely, whether the number of hops (unweighted) or the ease of information flow (weighted) is the more relevant feature in the context of the study. Furthermore, these metrics are often used as proxies for functional connectivity; that is network distance can be used as an indicator of how strongly distinct brain regions coactivate. Given this, it would be interesting in future studies to explore whether weighted or unweighted distances are better suited to capturing changes in structure-function relationships across the lifespan, and whether these align with the turning points identified by the authors. That said, I do not have a strong objection to the authors' choice; I would simply suggest that it be grounded in a more conceptual argument rather than in descriptive differences alone.

Overall, I appreciate the authors' revision of the manuscript. I believe the paper has significantly improved in clarity and rigor, and I have no further concerns.

(Remarks on code availability)

Reviewer #2

(Remarks to the Author)

The authors have adequately addressed my previous concerns.

1. My original query in comment 3 also asked about the decision to average age-predicted metrics before the UMAP projection, which remains unclear. The manuscript states, "Manifolds were constructed using significant age-predicted metrics... which were averaged for each age," but does not explain why this aggregation was necessary. This age-averaging step appears to oversimplify the underlying population data by collapsing individual variability into a single mean trajectory. Could the authors clarify the specific rationale for performing age-averaging before applying UMAP rather than projecting individual-level GAM predictions directly, and how this choice might influence the resulting embedding?
2. Regarding the boxplot figure in Extended Data Figure 4b, I now understand that the issue was not merely an axis-scaling problem but that PC3 was accidentally plotted in place of both PC1 and PC2. The explanation that "the boxplots are too short to really see the differences" does not accurately describe the error. I have reviewed the GitHub code and can confirm that the figures are reproducible.

(Remarks on code availability)

REVIEWER COMMENTS

Reviewer #1 (Remarks to the Author):

Summary:

Mousley et al., describe trends of brain network topology across the human lifespan. Specifically, they used diffusion MRI data from several cohorts to construct structural brain networks spanning ages 0 to 90 years and then compute a range of graph-theoretical measures. Their findings reveal age-related trajectories in these metrics, and highlight four major turning points that delineate epochs of brain development and aging.

The scope of this paper is relevant to both network neuroscience and lifespan research. Furthermore, the authors' effort to integrate multiple datasets is commendable, and the analyses are generally rigorous and well presented. I am overall supportive of the work. However, there are a few points the authors should address to make the paper robust, accessible, and interpretable.

Major comments:

One of my initial observations is that the graph theoretical measures are introduced in a list-like manner, only broadly grouped into categories such as integration, segregation, and centrality indices. However, the manuscript would benefit from a more in-depth rationale for including each measure. First of all, what do integration, segregation, and centrality mean in the context of brain networks? Then, it would be helpful to briefly explain what each metric captures in terms of brain network organization and why it is important in the context of the lifespan. Maybe they have been associated with specific cognitive functions in the early lifespan, or are biomarkers of neurodegenerative disease? Providing this context would enhance the interpretability of the results and help readers understand the functional relevance of the observed trends.

We agree that a better explanation of the intricacies of integration, segregation, and centrality would help interpretability. We have added to the introduction to improve the broader context, as well as added a section to the results that more clearly illustrates what these measures mean in the context of brain networks.

Introduction lines 24-41: "Prior research has revealed significant differences in structural topology associated with both individual differences^{6,9-11} and lifespan development^{4,12-15}. A typically developing infant's brain network displays adult-like structure with hub distribution, rich clubs, small-worldness,

and modularity at birth^{16–25}. Throughout early development, networks become more integrated with increasing strength and efficiency and decreasing modularity^{26–28}. That is, the network is becoming generally more strongly connected. In adulthood, many researchers describe an inverted “U” shape of development with a peak occurring around 30 years old where the brain is maximally efficient and integrated^{13–15}. This research uses the terms inflection point^{1,2,29,30} or peak age^{14,15} to describe important points of change in organizational metrics – many of which occur in the fourth decade of life and intersect with other developmental and aging milestones. After this point and into late life, aging is associated with reduced connectivity, mainly through pruning of weak connections^{12,14,31}, increased modularity¹³, and more pronounced rich club organization¹⁴ than earlier in life. In addition to these age-related changes, topological variation is associated with differences in individual outcomes. For example, there is a positive association between global efficiency (more short paths for information transfer) and intelligence in children⁷ and negative association between global efficiency and cognitive impairments in aging individuals¹¹. These established variations and lifespan fluctuations of organizational principles underscore the dynamic and complex nature of topology development.”

Results lines 106-120: “Topological metrics can be categorized as measures of network integration, segregation or centrality. Integration measures, such as global efficiency, assess the ease of communication across the network³⁹. Highly integrated topology is typically achieved by the network being well-connected by short path lengths, which conveys that the network is optimized for efficient communication³⁹. Network segregation, on the other hand, relates to partitioning the network into subgroups (e.g., modules), which are typically measured through the density or strength of within-group connections³⁹. Segregated topology increases the network’s capacity for specialized processing by use of subunits within a larger complex network^{39,41}. Lastly, measures of centrality convey the presence of nodes that are particularly important for network function (i.e., ‘central’ nodes)³⁹. For example, a node that is a member of numerous shortest paths is highly central as it plays a key role in information transfer. Thus, centrality not only facilitates network communication but also increases networks’ resilience to random knockouts of nodes⁴². All these concepts are important to understand in the context of the lifespan because different topological structures have strengths and weaknesses related to network function and thus provide clues as to the ‘goals’ of developmental change.”

One recommendation, then, would be to validate the results against null models. Specifically, the authors could generate random networks that preserve key properties—such as density and degree distribution—and

compute the same graph-theoretical indices on these surrogate data. Comparing the observed trends to those derived from the null models would help establish whether the identified trends and turning points are statistically meaningful and not simply a byproduct of network size or structure. Incorporating null models would strengthen the robustness of the findings and support the interpretation that the reported age-related changes reflect biologically relevant patterns.

We agree with the benefits of using null models to validate results. In fact, we originally thought along exactly the same lines as the reviewer. However, we realized that the nulls do not provide much insight in this specific context because we are using density-controlled networks. We have constrained every network to 10% density; thus, our topological analysis already preserves density. Due to this choice, any topological differences we observe *cannot* be due to differences in network size.

Furthermore, due to this strict density conservation, age-linked changes in degree distributions capture relevant differences in organisational structure. For example, if we create randomized null networks with preserved degree distributions and explore global efficiency, we see that all but one inflection point disappear (Fig. 1a). While the majority of age-related changes in global efficiency do not appear in null models, we do observe a small peak around 30 years old (Fig. 1a). We believe this is explainable by the age-related differences in degree distributions (Fig. 1b). Based on three five-year age groups (5 and younger, 28 through 32 years, and 85 and older), we see that there is a difference among the high-degree nodes. The middle-aged group has the highest degree across the groups (34). Additionally, in the middle-aged group, there are 20 nodes that higher than the highest degree in the old group (Fig. 1b). Thus, this demonstrates that the age-linked degree distributions of these networks are important topological context. Around 30 years old, the network is more capable of higher levels of efficiency than at other points in the lifespan due to the prevalence of more high-degree nodes.

This principle stands for many of the topological metrics we explored, given that we are not looking at spatial layout (i.e., topography). Another example is the core/periphery structure, which assesses the segregation of the network into a dense core and a sparse periphery. When the degree distribution is fixed, even a completely random edge layout will retain much of this topological structure (Fig. 1c). The observed and null core/periphery structure are significantly correlated (Fig. 1c; $r = 0.57$, $p < 2.20 \times 10^{-16}$). We believe this correlation is explainable because we are examining global topology, not topography (i.e., our analysis does not look at the spatial layout of the organisation). Even if the edges are put in different locations, global metrics such as core/periphery structure will still be highly similar because *where* those connections are is not relevant.

Overall, we completely agree that using null models to validate results is often a beneficial method. However, in this context, given that we constrained density and are exploring general topology, we don't believe it provides a basis for assessing the confidence of our results.

Figure 1. Exploration of null models. (a) Changes in global efficiency across age in observed (grey, left y-axis) and null (green, right y-axis) networks. Both regressions peak around 30 years old. (b) The degree distributions of three age groups – young (5 and younger), middle (28 through 32 years) and old (85 and older). (c) A significant correlation exists between observed and null network core/periphery structure.

Regarding the more operational aspects of the projects, three points require clarification.

1. The method for pruning connections to correct for density is unclear. Specifically, it is not stated which connections were retained—were they the strongest connections or the most frequently observed across the population or age bins? The function `threshold_absolute` referenced in the GitHub repository appears to be missing, and the text does not explain it clearly. Both approaches are justifiable: retaining the strongest connections highlights those most reliably identified by tractography, whereas preserving the most common and maybe also weaker connections acknowledges their documented presence in the brain and their role in linking distant brain regions [1] and promoting inter-areal diversity in connectivity profiles [2]. Regardless of the chosen method, it should be clearly stated and justified in the text to enhance transparency and reproducibility.

We apologize for the lack of clarity on thresholding. The reason `threshold_absolute` is not in our repository is because it is a Brain Connectivity Toolbox function. We have updated the comments in our code to clarify where this function is coming from. The threshold method is an absolute cut-off based on weight, which in our networks is the number of streamlines. Thus, we thresholded by retaining the strongest connections in the network. We have now clarified this in numerous locations:

Results line 88-91: “Before exploring network organization, we first examined general changes in connectivity across the lifespan by preserving the distribution of density by applying an absolute streamline-count threshold that yielded densities that were 70% of the average raw density per single-year age bin.”

Methods line 618-620: “Then, for each age group within each study, we applied the absolute threshold based on a streamline count cut-off that yielded an average density equal to the target density for that age.”

Methods line 624-626: “These networks were thresholded by the streamline counts so that every individual, regardless of age, had a 10% dense network.”

2. The authors computed modularity using the function `modularity_und`, providing only the adjacency matrix as input. This choice effectively sets the resolution parameter to its default value ($\gamma=1$) and thus constrains the analysis to a single spatial resolution. While this approach is sometimes used in the network neuroscience literature, it is well established that brain networks exhibit modular organization at multiple scales [3,4], often organized in a hierarchical manner [5]. Best practices recommend to identify modules and modularity by sweeping the γ parameter across a range of values to fully characterize modular organization. Therefore, the authors should clearly state that their results are based on a single partition and resolution, justify this choice, or present it as a limitation. A relevant reference for the paper in general and specifically in this context is [6], where age-related trends of modularity have been analyzed across the lifespan at a single scale, and the analysis was replicated at different spatial resolutions in the supplementary material. Additionally, the way modularity is described in the text is not technically accurate (line 110 page 5). Modularity is not defined as “the division into non-overlapping, highly-interconnected node groups”. Rather, it is a quality function that quantifies how well a network can be partitioned into such groups. This distinction is important for correctly interpreting the metric.

We agree that our use of one resolution for modular analysis is a limitation. Our choice was made for the purpose of simplicity and clarity, in support of our overarching goal of identifying topological turning points by incorporating 12 different topological measures. However, we agree that including some optimization procedure for choosing γ is better practice. We have redone the analysis and swept through 10 γ values ranging from 0.2-2 by 0.2 increments (Fig. 2). We repeated this procedure with randomized networks with preserved degree distributions. Using the Kolmogorov-Smirnov statistic to compare the distributions of modularity, we found significant differences in modularity between observed and randomized networks at every level of γ (Fig. 2a; all $p < 0.001$). The difference between observed and random network distributions was most distinct at $\gamma = 0.6$

(Fig. 2a; $KS = 0.93$). Thus, at this spatial resolution, modularity is the most non-random and therefore we have updated our analysis to include modularity assessed at $\gamma = 0.6$. Notably, the lifespan trajectories of maximum modularity are highly consistent in terms of inflection points (Fig. 2b). Sweeping the gamma parameter affected the overall magnitude of maximum modularity, not at what ages peaks/valleys in modularity occur (Fig. 2b). We integrated this analysis into the manuscript:

Results lines 152-154: “Modularity, how well a network can be divided into non-overlapping, highly intra-connected node groups⁴⁶, significantly fluctuated across the lifespan with a minimum at 31 years old and a maximum at 90 years old (Fig. 2b; Table 1).”

Methods lines 638-643: “Modularity was calculated at one spatial resolution ($\gamma = 0.6$), which was chosen after sweeping through gamma values from 0.2 to 2. At each level, observed modularity was compared to the modularity of randomized networks with preserved density and degree distributions. The spatial resolution was decided based on which level had the largest Kolmogorov-Smirnov (KS) statistic⁹¹, indicating that the modularity structure was the most non-random (Supplementary Fig. 9).”

Supplementary Figure 9. Modularity at varying spatial resolutions. (a) The distributions of modularity at differing spatial resolutions (gamma) are compared between observed and randomized networks (with preserved density and degree distributions). KS is the Kolmogorov-Smirnov statistic, and all these distributions are significantly different ($p < 0.001$). (b) Generalized additive models of modularity across age at each level of gamma highlight that the location of inflection points is highly consistent across spatial resolution.

3. In the script D_calculate_organization_measure.m, the characteristic path length and global efficiency are computed based on a connection length

matrix “dist”, which is a transformation of the structural connectivity matrix. However, the transformation is performed using the function `distance_bin.m`, instead of its weighted equivalent for undirected networks (`distance_wei.m`). Could the authors clarify this choice?

We are happy to clarify as we believe there is an argument for using either `distance_bin` or `distance_wei`. Ultimately, we used `distance_bin` because it retained more variability across networks compared to `distance_wei` for our data. For binarized distance, we get distances with a mean of 2.49 and a standard deviation of 0.87. On the other hand, when using weighted distance, we get much smaller, less varied distances ($M = 0.08$, $SD = 0.05$). We believe this is due to our use of *normalized* weighted networks. These networks have edge weights constrained to 0-1, which is heavily right-skewed ($M = 0.08$, $SD = 0.11$). Due to this distribution, using weighted distance results in extremely small path lengths compared to using binarized distance. Thus, in this scenario the weights do not appear provide a better estimation of path lengths and therefore, we choice to use binarized distance. Additionally, the *relative* distances across the networks are highly similar between binarized and weighted distance calculations (Fig. 3). This is important as it suggests that by choosing either method we are not influencing regional relationships as would be the case if two nodes were connected by a short binarized distance but large weight distances or vice versa.

Figure 3. Average distance matrices. (a) Binarized and (b) weighted distance matrices suggest similar relative distances across networks but at different scales.

I found inconsistencies in the descriptions of age-related trends in the 5 epochs.

Epoch 1: 0-8 years old. The authors say that there is a decrease in small-worldness (line 200), whereas instead it increases in this age range.

Epoch 2: 8-32 years old. The authors first say that there is a decrease of network integration (lines 219,220) and then they say the opposite (line 225). I believe global integration increases in this epoch. Moreover, at line 235 they

state “betweenness centrality happens around 32 years old”. What does it mean?

We apologize for these inconsistencies, which were indeed in-text errors. The issue with betweenness centrality on line 235 was due to a grammatical error with a sentence fragment. This sentence was aimed to communicate that both modularity and betweenness centrality are decreasing before but increasing after 32 years old. We have corrected these errors in-text:

Results lines 221-223: “Thus, while an increase in small-worldness across this period is the largest *directional* pattern, the local-level clustering coefficient is the crucial predictor of age (Fig. 4a).”

Results line 242-244: “Within this epoch, all topological measures were significantly correlated with age, characterized by increasing network integration and complex segregation and centrality patterns (Fig. 4e; Table 2).”

Results lines 257-259: “This result suggests a shift around 32 years old from increasing to decreasing integration as well as changes from decreasing to increasing modularity and betweenness centrality.”

Minor comments:

It would be interesting (maybe as a supplemental analysis) to explore the regional patterns underlying the local graph-theoretical indices. While their average is used to determine turning points of the lifespan, a local analysis could reveal which specific brain areas are driving these trends within each of the five epochs. For example, what are the brain regions majorly driving the decline of betweenness centrality in the second epoch? Are these brain regions also driving the subsequent rise of that metric?

While local-level organization was not a part of our original goal of the project, we agree that it is an interesting addition and, importantly, may be beneficial to readers with a specific interest in topography. We have added a supplementary analysis and for the instances where the local measures were the highest correlation or largest predictor of age (LASSO), included simple summaries in the main manuscript.

Results lines 213-214: “Local-level correlations between measures and age within each epoch are in the supplement (Supplementary Fig. 6).”

Results lines 223-225: “Significant correlations between clustering and age were found in 55 out of the 90 regions after false-discovery rate (FDR) correction, and these regions were disrupted across the brain (Supplementary Fig. 6c).”

Results lines 271-274: “Clustering coefficient was significant in 71 regions while local efficiency was significant in 74 regions (after FDR correction), indicating this relationship was disrupted across the majority of the brain (Supplementary Fig. 6a,c).”

Results lines 316-319: “Importantly, subgraph centrality was only significantly correlated with age in 10 regions (Supplementary Fig. 6e), including the cuneus (right and left), the superior (right) and middle (left) occipital gyri, and the postcentral gyrus (right). Thus, an increase in centrality in late life has a spatial-temporal pattern.”

Methods lines 636-638: “All local measures were averaged across the network for the topological analysis, though local-level correlations between measures and age are in the supplement (Supplementary Fig. 6).”

Methods lines 740-742: “We also provided local-level correlations between measures and age within each brain region after false-discovery rate (FDR) correction⁵ (Supplementary Fig. 6).”

Supplementary Figure 6. Local correlations between organisational measures and age within each epoch. (a) Local efficiency was significant in 61 regions from 0-9 years old (epoch one), 70 regions from 9-32 years old (epoch two), 74 regions from 32-66 years old (epoch three), four regions from 66-83 years old (epoch four), and one region from 83-90 years old (epoch five). (b) Strength was significant in 31 regions in epoch one, 68 regions in epoch two, 71 regions in epoch three, eight regions in epoch four, and no regions in epoch five. (c) Clustering coefficient was significant in 55 regions in epoch one, 65 regions in epoch two, 71 regions in epoch three, nine regions in epoch four, and one region in epoch five. (d) Betweenness centrality was significant in 22 regions in epoch one, 50 regions in epoch two, 30 regions in epoch three, two regions in epoch four, and no regions in epoch five. (e) Subgraph centrality was significant in 19 regions in epoch one, 63 regions in epoch two, 35 regions in epoch three, 10 regions in epoch four, and 10 regions in epoch five. Values shown are the r values of all significant correlations after FDR correction. The surface plots show r values for each epoch.

At line 85 (page 5) and 561 (page 23) the authors mention age bins. How are they defined? Do they comprise a single year of the lifespan or multiple?

We apologize for not clearly defining age bins. The age bins are comprised of a single year. We have clarified this in-text:

Results line 88-91: “Before exploring network organization, we first examined general changes in connectivity across the lifespan by preserving the distribution of density by applying an absolute streamline-count threshold that yielded densities that were 70% of the average raw density per single-year age bin (Fig. 1c; Supplementary Fig. 1a).”

Methods line 610-613: “Before thresholding, 14 participants were identified as outliers (dHCP $n = 1$; CALM $n = 2$; RED $n = 1$; ACE $n = 1$; HCPya $n = 3$; HCPa $n = 1$; CamCAN $n = 5$) due to having network density above or below three standard deviations for the age bin (comprised of closest rounded year) and were removed.”

At line 355 the authors say “centrality is important during adolescence but minimally for the rest of the life”. I think this sentence is imprecise. Centrality is always important, maybe the authors mean that variations in centrality measures show the most prominent age-related trends during adolescence?

We agree that this line does not accurately represent changes in centrality metrics. We were indeed claiming that variations in centrality measures show the most prominent age-related trends during adolescence, that section has now been rewritten in the revised manuscript.

At line 378 the authors say “the second lifespan epoch, ages 8 to 20, indicates no significant shift in the trajectory topological development”, but didn’t they observe a shift in the trends of topological organization?

We apologize for the confusing phrasing. The line is referring to the second epoch, eight to 32, but is imprecisely describing the difference between turning points and general topological change. Meaning, topology does change across this period, but these changes occur along the same trajectory

(i.e., there are no major topological turning points within the age range). We have revised:

Discussion lines 414-415: “The second lifespan epoch, ages nine to 32, indicates that the trajectory of topological development remains consistent across this period.”

Lastly, the authors mention “extended data” (page 5 line 95), and “statistics” (page 23 line 565). However, at least to me, it is not clear where to find these sections. Are these files separate from the Supplementary Information?

We see how this was unclear. Line 95 is referring to supplementary information, which we have changed. Line 565 was referring to the statistics section within the methods, we have revised this as well to be clearer:

Results lines 103-106: “This method allows for fair comparison of topological structure across the lifespan without total connectivity biases, though a full topological analysis with variable density networks is also provided (Supplementary Fig. 2, Table 3).”

Methods lines 616-618: “For the variable density analysis, we performed a generalized additive model (see “Methods”, “Statistics”) on the raw network densities and took 70% of the regression to obtain a ‘target’ density for each age (Supplementary Fig. 1a).”

References:

- [1] Distance-dependent consensus thresholds for generating group-representative structural brain networks, 2019, R.F. Betzel, Al. Griffa, P. Hagmann, B. Mišić, *Network Neuroscience*.
- [2] Specificity and robustness of long-distance connections in weighted, interareal connectomes, 2018, R.F. Betzel and D.S. Bassett, *PNAS*.
- [3] Multiscale Network Organization in the Human Brain, 2013, D. Bassett and F. Siebenhühner, in *Multiscale analysis and nonlinear dynamics*
- [4] Multiscale brain networks, 2017, R.F. Betzel and D.S. Bassett, *Neuroimage*
- [5] Hierarchical modularity in human brain functional networks, 2009, D. Meunier, R. Lambiotte, A. Fornito, K.D. Ersche, E.T. Bullmore, *Frontiers in Neuroinformatics*
- [6] The modular organization of brain cortical connectivity across the human lifespan, 2020, M.G. Puxeddu, J. Faskowitz, R.F. Betzel, M. Petti, L. Astolfi, O. Sporns, *NeuroImage*

Reviewer #1 (Remarks on code availability):

The results are mostly reproducible. There is one function that is missing (see comments for the authors)

Reviewer #2 (Remarks to the Author):

This paper investigates the non-linear development of structural brain topology across the human lifespan by analyzing diffusion imaging data from 4,216 participants aged 0–90 years. The authors compute 12 graph theory metrics to quantify network organization and then use Uniform Manifold Approximation and Projection to project these high-dimensional data into a lower-dimensional manifold space. By doing so, they identify four major topological “turning points” at approximately 8, 32, 62, and 85 years, which delineate five distinct epochs of brain network development. The study links these turning points to changes in integration, segregation, and centrality measures and discusses how these findings align with known neurobiological milestones and cognitive developmental phases. The study is novel and interesting, developing an innovative approach to mapping lifespan changes in brain network topology.

Specific comments

1. It is unclear whether the authors accounted for potential confounding variables, such as ethnicity, socioeconomic status, or other factors, in their analyses. Clarifying this would strengthen confidence in the generalizability of the reported findings.

We apologize that this was not clear. We took a population-representative approach and thus we did not control for specific variables (e.g., socioeconomic status) that vary across populations. We have added a section in the methods to clarify what was controlled for:

Methods lines 644-647: “All topological measures were assessed using generalized additive models. In these models, cubic regression splines were used to smooth across age, and sex, atlas, and dataset were controlled for.”

2. The manuscript effectively employs UMAP to identify turning points in brain network topology across the lifespan. To further assess the robustness of these findings, please comment on the sensitivity of the identified turning points to the choice of dimensionality reduction technique. For instance, it would be informative to discuss whether linear methods (e.g., PCA, PLS), which provide interpretable component loadings, might yield comparable turning points. Similarly, a discussion comparing UMAP’s results to those potentially obtained from other non-linear methods, such as diffusion map embedding, would be valuable. Additionally, please clarify the rationale for reducing the topological data specifically to three dimensions. Could exploring higher-dimensional UMAP embeddings potentially reveal finer-grained, yet

meaningful, epochs of topological development beyond the five major ones reported?

Thank you for the thoughtful questions! We decided not to go with linear methods such as PCA or PLS because we believe it's fundamentally inaccurate to assume that lifespan development will occur in a linear fashion. The goal of the project is to detect *changes* in developmental trajectories, not to identify single sources of linear development. Thus, whilst we do use linear methods in the manuscript (e.g. PCA), these are to interpret the periods between the turning points due to the benefit of having component loadings. In terms of non-linear dimensionality techniques (e.g., UMAP, t-SNE, and diffusion map embedding), we believe there is a rationale for using any of these methods and future work should do so to further test turning point reproducibility. The main reasons why we went with UMAP are due to it being fast to run (e.g., faster than t-SNE) (McInnes et al., 2018), which was an important consideration given that we have generated over 900 UMAPs with varying parameters. Additionally, the UMAP is particularly good at capturing both global and local patterns in the data (McInnes et al., 2018). We felt this was an appropriate method given that we want to capture a general landscape of topological relationships.

Additionally, we chose three dimensions for the UMAP because we thought that it was the highest dimensionality that would still be easily interpretable. The higher the dimensions, the harder it is to plot and analyse. As we have not seen others employing a UMAP in this way, we chose to err on the side of simplicity for the sake of validating that this methodology is informative. We think that future work should explore the potential existence of more granular turning points that could be discovered using more dimensions.

We have made in-text changes to clarify these points and highlight limitations:

Results lines 185-197: “Thus, we reduced the dimensionality of this data using manifold learning to examine non-linear changes in lifespan topology. These manifolds are 3-dimensional topological spaces that capture crucial patterns in the data.”

Results lines 331-334: “While the fact that PCA is a linear method makes it less suitable for identifying where fluctuations occur in the data compared to manifold learning techniques, it is a useful tool for comparing the pre-defined epochs as it generates interpretable loading scores.”

Discussion lines 484-489: “Additionally, our manifold spaces were constrained to three dimensions for straightforward interpretation; however, higher-dimensional UMAP embeddings could potentially reveal finer-grained but important turning points in topology. Finally, although we employed UMAP for its speed and its ability to capture both global and local structure in the data³⁵,

applying other non-linear techniques—such as diffusion map embedding or t-SNE—could offer valuable context for assessing the robustness of these results.”

Methods lines 658-661: “The number of components simply determines how many dimensions the projection should be embedded in. We predefined this as three dimensions in an effort to capture multi-dimensional changes without losing interpretability.”

3. The authors use polynomial fitting on age-averaged UMAP projections to identify turning points, and they have clearly made efforts to optimize the fitting parameters and smooth out small fluctuations. However, it is unclear why the same approach used to model the 12 topological metrics, namely, fitting GAMs to individual-level data, was not applied directly to the individual UMAP projections. Would this alternative approach yield similar turning points, and might it avoid potential information loss introduced by age-averaging? It would be helpful if the authors could justify their methodological choice and discuss whether the findings are robust across different modeling strategies.

We think applying GAMs themselves to the UMAP space is an interesting idea. When considering this choice, we decided to go with polynomial fits for a few reasons. Mainly, the data projected into the UMAP are the age-predicted results from the individual GAMs in which sex, atlas and dataset have been controlled for. This means that if we fit GAMs to the UMAP space, we are essentially fitting new GAMs based on the results of previous GAMs. We worried that this would not only be confusing but also would not be an appropriate use of the modelling method as the models would likely be highly over-fitted due to being run twice on the same data. Additionally, the benefit of GAMs is the ability to smooth across variables (i.e., age) while controlling for other metrics (i.e., sex, dataset and atlas). Since we have already done this, what the model is using to predict the regression is less clear. Do we use dimensions one and two to predict dimension three? Do you use the same smoothing parameters across all dimensions? We think applying the GAM method in this scenario isn't more informative because we aren't dealing with known additive conditions (e.g., age) and we aren't interested in controlling for certain variables, as this has already been done. Thus, we feel this is a good situation for simpler and clearer line fitting, which is why we use polynomial fits.

Overall, we believe an argument could be made to use any line-fitting methods. We decided to use a polynomial fit to keep interpretation as simple as possible and avoid potential complications with fitting GAMs on GAM-predicted data.

4. This paper effectively delineates key topological transitions across the human lifespan. The authors employed three distinct methodological approaches (Pearson's correlation, Lasso regression, and Principal Component Analysis) to characterize the key metrics at each stage of structural development. However, the Discussion section would be strengthened by a more comprehensive interpretation of the neurobiological basis of these graph metrics. It is suggested that the authors expand the manuscript to consider how the key metrics identified within and between epochs may relate to specific white matter maturation or degeneration, and the contribution of these processes to the observed lifespan epochs.

This is a great point. We agree that it is important to bridge the gap between topology and structural changes across the lifespan. We have expanded our discussion to include more details on the relationship between white matter and topological development:

Discussion lines 399-406: "The first few years of life are marked by consolidation and competitive elimination of synapses²⁰ and rapid increases in gray and white matter volume¹. Our results indicated that, topologically, structural networks develop along the same dimensions from birth until about nine years old. This is consistent with a previously identified cortical turning point around seven years old, where global efficiency reaches a minimum, cortical thickness peaks, and cortical folding stabilizes⁵⁰. Thus, within the first decade of life, while myelination and white matter volume increase rapidly, topological efficiency decreases parallel to synaptic elimination."

Discussion lines 424-431: "These findings are highly consistent with previous work exploring individual topological metrics¹³⁻¹⁵ that identify significant peak/inflection points at the beginning of the fourth decade. Beyond organizational changes, this turning point aligns with developmental trajectories of white matter. White matter volume and fractional anisotropy peak around 29 years old^{1,64,65}, mean diffusivity arrives at a minimum around 36 years old^{64,65}, and radial diffusivity reaches a minimum around 31 years old⁶⁴. Thus, across the phase of life, while white matter integrity and volume are increasing rapidly, topological structure at the macroscale is becoming more efficient and less segregated."

Discussion lines 435-439: "Compared to rapid maturation in earlier life, changes in network architecture slow during this period^{40,64,65}, which is consistent with our results that there are no major topological turning points until the 60s. Aligned with the slowing of white matter maturation during this period, the patterns of topological change are less complex than previous epochs, with clear increases in segregation and decreases in efficiency."

Discussion lines 447-453: "Therefore, this turning point may reflect protracted or accelerated development. Indeed, accelerated decreases in white matter

integrity are known to occur in late life⁶⁴. This decrease in white matter integrity is generally referred to as 'age-related' degeneration, meaning reductions in white matter coherence are expected in late aging individuals^{3,66}. Topologically, we find that during this phase macroscale reorganizational patterns simply – with the most distinct change being increasing modularity^{13,67,68}. Together, these patterns suggest a sparsification of the structural network in aging.”

5. In the Discussion, the authors interpret the decline in significant age–topology correlations across later life epochs as evidence of a weakening relationship between age and structural brain topology. However, could this pattern be confounded by reduced sample sizes in older age groups (e.g., from age 30s to 90s)? Clarifying whether statistical power was considered in this interpretation would strengthen the argument.

We agree that this is an important thing to point out to the reader. We did include a reference to this confound in the discussion, stating that the weakening relationship between age and structural topology could be due to sample size differences. However, to clarify this further, we have added the mean statistical power of correlations within each epoch. We added this to the results and discussion:

Results lines 311-313: “Importantly, compared to all epochs, this epoch has the lowest statistical power due to sample size (mean power for epoch one = 0.72, epoch two = 0.97, epoch three = 0.92, epoch four = 0.35 and epoch five = 0.16).”

Discussion lines 460-462: “It is possible that the lack of significant findings reflects the small sample size ($n = 93$), which is reflected in the low statistical power in this epoch.”

6. In Extended Data Figure 4b, the three boxplots appear identical, despite differing significance indicators. Could the authors double-check the accuracy of this figure and verify that all visualisations throughout the manuscript correctly reflect the underlying data?

Thank you for pointing this out! We agree that the boxplots appear similar, making it hard to interpret. The data represented is correct, however, the boxplots are too short to really see the differences. We have edited these figures by extending the y-axis to better depict significant differences. We have verified other figures as well and will include source data in the submission so that figures can be replicated.

Supplementary Figure 7. Additional visualizations of epoch-based analyses. (a) Spider plots of Pearson's r values for each graph theory measure in consecutive epochs highlight the epochs where the direction of significant age-topology relationship (indicated by red highlighted measures). (b) PCA scores per epoch across each PC. Across PC 1 scores, epochs one and two ($p = 0.002$), two and three ($p = 4.95 \times 10^{-13}$), and three and four ($p = 1.01 \times 10^{-14}$) are all significantly different. Similarly, for PC 2 scores, epochs one and two ($p = 2.15 \times 10^{-10}$), two and three ($p = 0.002$), and three and four ($p = 2.47 \times 10^{-13}$). Lastly, epochs three and four ($p = 1.82 \times 10^{-13}$) significantly differ across PC 3 scores, while epochs four and five only significantly differ in PC 2 scores ($p = 0.008$). (c) PC score series were created from average score for each age. These series were used in the Dynamic Time Warping analysis to explore the shape of trajectories. *** indicates $p < 0.001$, ** indicates $p < 0.01$, * indicates $p < 0.05$.

REVIEWERS' COMMENTS

Reviewer #1 (Remarks to the Author):

The authors have responded to all the issues raised in the initial review, and I appreciate the additional analyses they have carried out to address my comments. I find the revised manuscript suitable for publication.

There is one aspect where our perspectives do not fully converge: the choice to compute network distances on the binarized matrix. Based on the response, it appears that this decision was made a posteriori, after observing the dispersion of distance values from both weighted and unweighted graphs. In my view, this is not a fully convincing rationale. Ideally, such a methodological choice should be guided by a theoretical hypothesis—namely, whether the number of hops (unweighted) or the ease of information flow (weighted) is the more relevant feature in the context of the study. Furthermore, these metrics are often used as proxies for functional connectivity; that is network distance can be used as an indicator of how strongly distinct brain regions coactivate. Given this, it would be interesting in future studies to explore whether weighted or unweighted distances are better suited to capturing changes in structure-function relationships across the lifespan, and whether these align with the turning points identified by the authors. That said, I do not have a strong objection to the authors' choice; I would simply suggest that it be grounded in a more conceptual argument rather than in descriptive differences alone.

Overall, I appreciate the authors' revision of the manuscript. I believe the paper has significantly improved in clarity and rigor, and I have no further concerns.

We appreciate you taking the time to explain your rationale for choosing the distance metric and understand your concerns about the posteriori decision. We agree that there is an important theoretical distinction between the metrics. We will keep this in mind for future projects! Thank you for your time and support of the manuscript.

Reviewer #2 (Remarks to the Author):

The authors have adequately addressed my previous concerns.

1. My original query in comment 3 also asked about the decision to average age-predicted metrics before the UMAP projection, which remains unclear. The manuscript states, "Manifolds were constructed using significant age-predicted metrics... which were averaged for each age," but does not explain why this aggregation was necessary. This age-averaging step appears to oversimplify the underlying population data by collapsing individual variability into a single mean trajectory. Could the authors clarify the specific rationale for performing age-averaging before applying UMAP rather than projecting individual-level GAM predictions directly, and how this choice might influence the resulting embedding?

We agree this is an important point and apologize that our rationale for averaging was not clearly explained. Our primary goal in this study was to identify *normative and generalizable turning points* across the lifespan, rather than to characterize individual variability. Given that our sample of combined datasets is cross-sectional, we do not have repeated scans that would enable to examine individual developmental trajectories. Thus, projecting individual-level GAM predictions would risk overinterpreting patterns as if they reflected longitudinal

changes, which our data cannot support. Thus, our decision to age-average was in part for conceptual clarity as it emphasizes that the data represented is a sample-wide trajectory and reduces the risk of misinterpretation that the results provide information about individual lifespan development. Additionally, age-averaging had the practical benefit of reducing the computational expense of our over 900 UMAP embeddings by requiring 90 datapoints be projected rather than over 3,000 individual datapoints.

We agree that aggregation inevitably collapses variability and leads to a loss of information. This is an inherent limitation of our cross-sectional project even though the aim of the project was to establish a normative trajectory. We believe that future work with longitudinal datasets capable of generating individual-level developmental trajectories will be critical to examine whether the topology and timing of turning points we observed are consistent across individuals.

2. Regarding the boxplot figure in Extended Data Figure 4b, I now understand that the issue was not merely an axis-scaling problem but that PC3 was accidentally plotted in place of both PC1 and PC2. The explanation that “the boxplots are too short to really see the differences” does not accurately describe the error. I have reviewed the GitHub code and can confirm that the figures are reproducible.

We apologize for the accidental repeat of PC3 and are glad to hear that the figures are reproducible.